# MeV-Stealth: A CD46-specific oncolytic measles virus resistant to neutralization by measles-immune human serum

**Miguel Ángel Muñoz-Alía** [1]*, **Rebecca A. Nace**[1], **Alexander Tischer** [2], **Lianwen Zhang**[1], **Eugene S. Bah**[3], **Matthew Auton** [2], **Stephen J. Russell**[1,2]*

**1** Department of Molecular Medicine, Mayo Clinic, Rochester, Minnesota, United States of America,
**2** Division of Hematology, Mayo Clinic, Rochester, Minnesota, United States of America, **3** Mayo Clinic Graduate School of Biomedical Sciences, Mayo Clinic College of Medicine and Science, Rochester, Minnesota, United States of America

* alia.miguel@mayo.edu (MÁM-A); sjr@mayo.edu (SJR)

**Data Availability Statement:** All relevant data are within the manuscript and its Supporting Information files.

## Abstract

The frequent overexpression of CD46 in malignant tumors has provided a basis to use vaccine-lineage measles virus (MeV) as an oncolytic virotherapy platform. However, widespread measles seropositivity limits the systemic deployment of oncolytic MeV for the treatment of metastatic neoplasia. Here, we report the development of MeV-Stealth, a modified vaccine MeV strain that exhibits oncolytic properties and escapes antimeasles antibodies *in vivo*. We engineered this virus using homologous envelope glycoproteins from the closely-related but serologically non-cross reactive canine distemper virus (CDV). By fusing a high-affinity CD46 specific single-chain antibody fragment (scFv) to the CDV-Hemagglutinin (H), ablating its tropism for human nectin-4 and modifying the CDV-Fusion (F) signal peptide we achieved efficient retargeting to CD46. A receptor binding affinity of ~20 nM was required to trigger CD46-dependent intercellular fusion at levels comparable to the original MeV H/F complex and to achieve similar antitumor efficacy in myeloma and ovarian tumor-bearing mice models. In mice passively immunized with measles-immune serum, treatment of ovarian tumors with MeV-Stealth significantly increased overall survival compared with treatment with vaccine-lineage MeV. Our results show that MeV-Stealth effectively targets and lyses CD46-expressing cancer cells in mouse models of ovarian cancer and myeloma, and evades inhibition by human measles-immune serum. MeV-Stealth could therefore represent a strong alternative to current oncolytic MeV strains for treatment of measles-immune cancer patients.

## Author summary

Vaccine strains of the measles virus (MeV) have been shown to be promising anti-cancer agents because of the frequent overexpression of the host-cell receptor CD46 in human malignancies. However, anti-MeV antibodies in the human population severely restrict the use of MeV as an oncolytic agent. Here, we engineered a neutralization-resistant MeV

**Funding:** This work was funded by grants from Al and Mary Agnes McQuinn and Mayo Clinic. The funders had no role in study design, data collection and interpretation, or the decision to submit the work for publication.

**Competing interests:** I have read the journal's policy and the authors of this manuscript have the following competing interests: The work described here is subjected to a patent application by the Mayo Clinic entitled "Engineering the Hemaglutinin and Fusion Proteins of Canine Distemper Virus" that has been out-licensed. MAM-A and SJR are named inventors. SJR is a founder and equity holder of Vyriad. MAM-A reports honorarium fees from Biomere. The other authors have nothing to declare.

vaccine, MeV-Stealth, by replacing its envelope glycoproteins with receptor-targeted glycoproteins from wild-type canine distemper virus. By fully-retargeting the new envelope to the receptor CD46, we found that in mouse models of ovarian cancer and myeloma MeV-Stealth displayed oncolytic properties similar to the parental MeV vaccine. Furthermore, we found that passive immunization with measles-immune human serum did not eliminate the oncolytic potency of the MeV-Stealth, whereas it did destroy the potency of the parental MeV strain. The virus we here report may be considered a suitable oncolytic agent for the treatment of MeV-immune patients.

## Introduction

Oncolytic viruses (OVs) are a clinically proven anticancer immunotherapy that can cause tumor debulking by selective killing of tumor cells (known as oncolysis) [1]. The vaccine lineage of the measles virus (MeV) is one of the most extensively investigated OVs that targets tumor cells by selectively binding to the membrane cofactor protein CD46 (MCP/CD46), frequently overexpressed in malignant tumors (https://www.proteinatlas.org/ENSG00000117335-CD46/pathology). MeV is an enveloped, negative-stranded morbillivirus that enters cells through fusion with the cell membrane mediated by two viral surface proteins, namely, the hemagglutinin (H) and fusion (F) proteins. Upon binding of the H glycoprotein to its cellular receptor, F-mediated membrane fusion is triggered. While the cellular receptors for pathogenic MeV are SLAMF1 (CD150) on immune cells and nectin-4 (PVRL4) on epithelial cells [2], vaccine or laboratory-adapted strains of MeV have acquired the ability to use CD46 as a receptor.

CD46 expression is associated with poor prognosis for tumors originating from the breast or female genital tract [3,4]. Furthermore, in hematologic malignancies, CD46 antigen density is 2–8 times higher in neoplastic cells than in their normal counterparts [5,6]. Our group, as well as others, has shown a strong correlation between CD46 expression and the oncolytic potency of vaccine-lineage MeV [5,6], further underscoring the usefulness of this cell-surface protein as a target for OVs. Thus, vaccine-lineage MeV is currently being evaluated in clinical trials as a CD46-targeted therapeutic agent for cancer treatment.

Despite the clinical promise of this treatment modality [7], preexisting immunity to MeV antigens can significantly inhibit direct oncolysis [8]. Indeed, the complete response of a multiple myeloma patient following a single intravenous infusion of MeV at the Mayo Clinic was associated with a lack of detectable neutralizing anti-measles antibodies at baseline and a high baseline frequency of measles-reactive T cells [9]. Given the widespread immunity to MeV in the general population and the desirability of systemic administration as an approach for the treatment of metastatic cancer there may be significant advantages if this viral platform could be engineered to evade serum neutralization.

Canine Distemper Virus (CDV) is a member of the genus *morbillivirus* that infects most carnivorous species [10], and shares high structural similarities with MeV [11,12]. Because of the low to absent levels of intergenus cross-neutralizing antibodies [13–15], exchange of the MeV envelope glycoproteins with those of CDV has been pursued as a strategy to minimize the effect of the widespread immunity against MeV [16]. However, clinical translation requires fully retargeted capabilities towards tumor-selective receptors [17].

In the current study, we sought to determine whether a fully attenuated MeV strain pseudotyped and CD46-targeted via engineered expression of modified CDV glycoproteins would have oncolytic efficacy equal to that of the current MeV agent. With ovarian cancer and

myeloma as our targets, we ablated known receptor tropisms from CDV envelope glycoproteins and engineered them to display a single-chain variable fragment (scFv) with specificity for CD46. To determine whether differences in the binding affinity for CD46 could affect the results, a panel of scFvs with different affinities for CD46 was displayed and the fusogenic properties of the respective viruses were compared. The virus with CD46-directed fusogenic potency similar to unmodified MeV-H was used for pseudotyping and comparison purposes, thus generating MeV-Stealth. The oncolytic properties of MeV-Stealth in mouse models of ovarian cancer and myeloma were indistinguishable from those of the parental MeV strain. In addition, passive immunization with measles-immune human serum did not eliminate the oncolytic potency of the MeV-Stealth, unlike the parental MeV strain. Thus, MeV-Stealth outperforms vaccine-lineage MeV in the setting of measles immunity. According to our results, MeV-Stealth may be suitable for clinical translation when systemic administration is planned.

## Results

### Heterologous combinations of wild-type CDV glycoproteins result in enhanced cell membrane fusion

We first sought to replace the MeV coat with an alternate viral coat that would enable the virus to evade neutralization by anti-measles antibodies. To do this, we selected wild-type CDV because (i) a detailed understanding of the molecular interaction of CDV-H with the natural receptors SLAMF1 and nectin-4 exists [18,19] and (ii) the phylogenetic proximity of the CDV envelope to that of MeV was deemed close enough to allow the CDV-H and CDV-F glycoproteins to be incorporated easily into the MeV structure (pseudotyping), generating viable virus that would avoid cross-reactive anti-measles antibodies [13,14,20]. While a strain of CDV approved for vaccine use exists (the Onderstepoort strain, OL), this strain can use a currently unidentified receptor in addition to SLAMF1 and nectin-4 (Fig 1), which would have made it challenging to modify the viral tropism. Therefore, we focused on wild-type strains as they are known to interact exclusively with SLAMF1 and nectin-4.

We next sought to identify the most fusogenic CDV-H/F glycoprotein pair. To do this, we transiently expressed different H/F combinations from the 5804P and SPA.Madrid/16 (hereafter named 5804 and SPA, respectively) isolates in Vero cells expressing either SLAMF1 or nectin-4 and assessed qualitatively the degree of cellular fusion (syncitia formation) induced by the viral proteins. Fusion activity for the CDV-H/F pairs was not observed when the 135 aa-signal peptide was maintained in CDV-F (Fig 2A). When swapped by the homologous from MeV F, coexpression of H/F proteins from 5804P resulted in cell fusion in SLAMF1- and nectin-4-expressing cells whereas coexpression of those from SPA promoted cell fusion exclusively in SLAMF1-expressing Vero cells. On the other hand, coexpression of the heterologous combination CDV-H 5804 and CDV-F SPA but not CDV-H SPA and CDV-F 5804 resulted in syncytia formation in SLAMF1- and nectin-4-expressing Vero cells.

The data summarized above provided evidence in support of a fusion defect on CDV-H SPA. To begin to address whether the lack of fusion of CDV-H SPA on nectin-4-expressing cells was due to a lower affinity for the receptor, we quantitatively compared fusion phenotypes via nonnatural receptors thus leveling the playing field for receptor binding affinity. Our approach was to fuse a HIS-tagged CD38-specific scFv to the C-terminal domain of the receptor binding proteins and to determine fusion levels in CHO cells encoding either CD38 or the pseudoreceptor for the HIS-tag (CHO-αHIS). We included into CDV-H SPA a L437M substitution since L437 corresponded to a clone-specific amino acid change not present in any other CDV genetic group (S1 Fig). For comparison purposes, we additionally included retargeted receptor binding proteins from other viruses: MeV-H and Nipah-G [21,22]. For added rigor

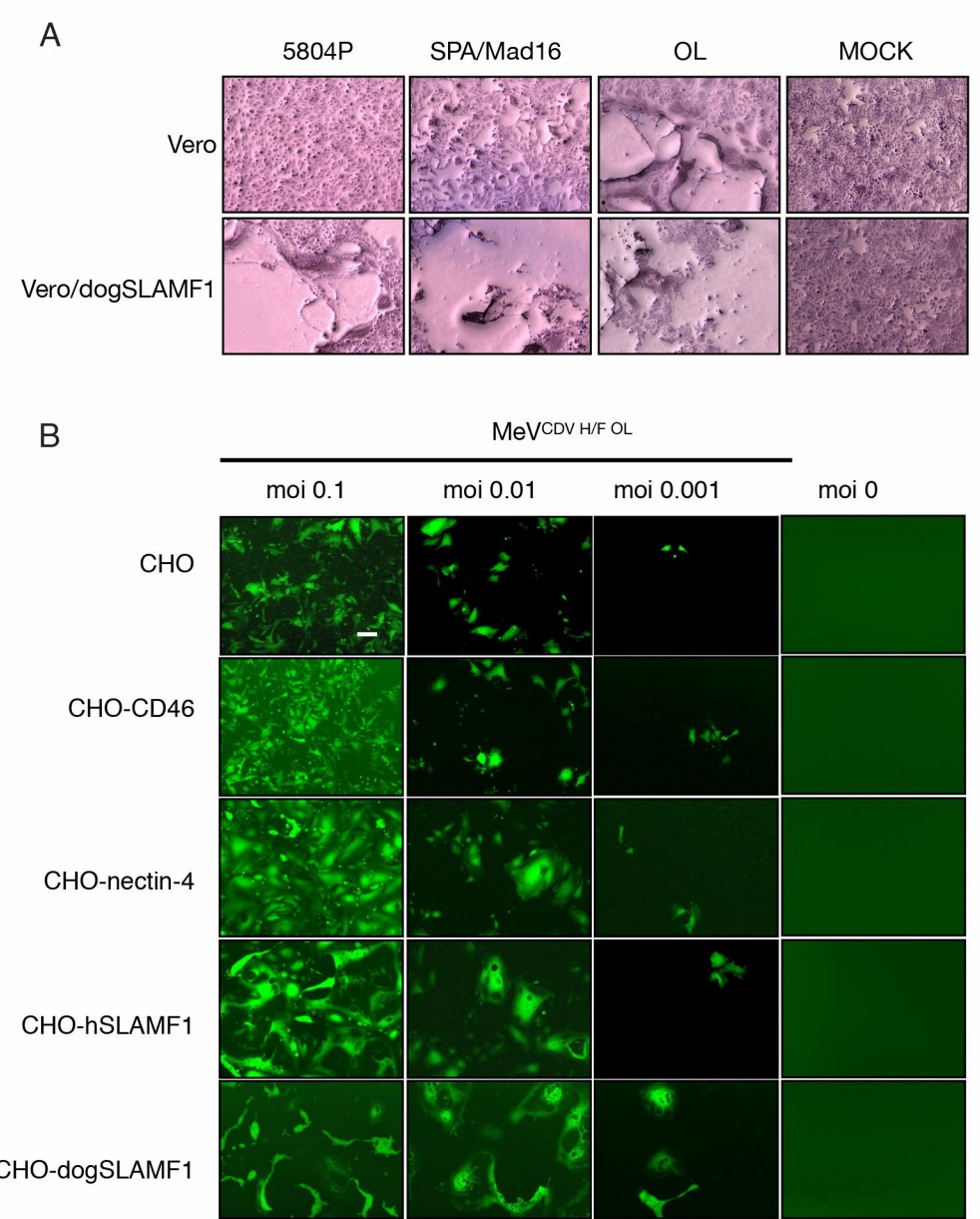

**Fig 1. CDV OL can infect cells lacking SLAMF1 and nectin-4 receptors. (A)** Assessment of infection of Vero cells by CDV isolates in comparison with the OL strain. Cells were infected at an MOI of 0.1 (determined on Vero-dogSLAMF1 cells) and stained with Hema-Quick 48 hours later for visualization. **(B)** A panel of CHO cells expressing different relevant receptors was infected with an eGFP reporter MeV comprising the CDV H/F OL glycoproteins. Infectivity was reported using a fluorescence microscope. Magnification 40X.

to our comparisons, expression of the receptor binding proteins was initially analyzed by western blot and flow cytometric analysis, which demonstrated no significant impact on protein folding or surface expression (S2A and S2B Fig, S1 Data). When the CDV-H/F pairs from SPA proteins were expressed in CHO-C38, only the CDV-H SPA with M437L promoted fusion (Fig 2B, S1 Data). Notably, no significant difference in fusion competence was observed between the two homotypic CDV-H/F pairs from the SPA or 5804P isolates. Surprisingly, the

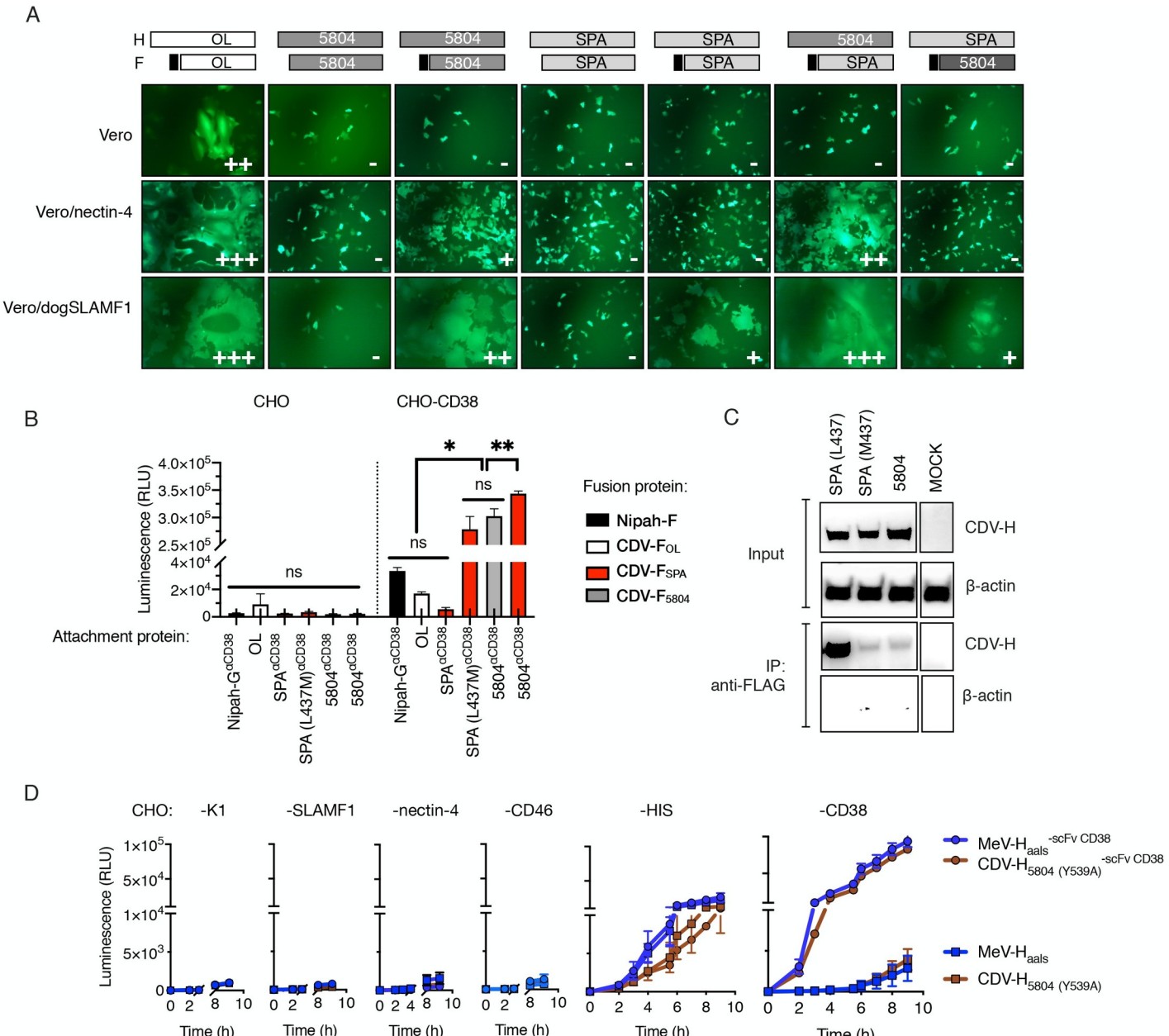

**Fig 2. Heterologous combination of wild-type CDV H/F with a shortened signal peptide results in enhanced receptor-dependent fusion due to a weaker H-F interaction.** (A) Syncytia formation in cells cotransfected with CDV-F, CDV-H and eGFP. The signal peptides of CDV-F were replaced with the homolog from MeV-F, as indicated by the black boxes in the schematic. Twenty-four hours after cotransfection, fusion score was assessed under the GFP channel. (B) Quantitative fusion assay. Effector BHK cells were transfected with the indicated combination of attachment (CDV-H or Nipah-G) and fusion (F) proteins plus one of the dual-split reporter plasmids. Target CHO cells and CHO cells expressing CD38 (CHO-CD38) were transfected with the other dual-split reporter plasmids. 16 hours post-transfection cells were overlaid and *Renilla* luciferase activity was determined (RLU) 8 hours later. Values represent the mean ± standard deviation (SD) of one representative experiment performed in triplicate. Statistical significance was determined using one-way ANOVA with Holm-Sidak's multiple comparison test (ns, not significant *, $p < 0.05$; **, $p < 0.002$; ***, $p < 0.0001$). (C) CDV-H/F coimmunoprecipitation. HEK293T cells transiently expressing either wt or mutant HIS-tagged CDV-H proteins together with FLAG-tagged CDV-F proteins were lysed and immunoprecipitated (IP) with an anti-FLAG antibody. The signal intensity was determined using an anti-HIS antibody. (D) Quantitative fusion assay of fully retargeted CDV-H and MeV-H proteins onto CHO cells and CHO cells derivates. Either HIS-tagged or HIS-tagged and CD38-retargeted MeV-H/F and CDV-H/F complexes were transfected into effector cells and luminescence signal was determined over time. MeV-Haals = MeV-H blinded for CD46, nectin-4 and SLAMF1 via amino acid substitutions Y481A, R533A, S548L and F549S.

fusion activity triggered by the heterologous H/F combination CDV-H 5804/F SPA surpassed that achieved by the homologous combinations CDV-H/F 5804 and CDV-H/F SPA. For the retargeted Nipah G/F pair, although we observed significant fusion levels in CHO-CD38 cells (p<0.0001), they were insignificant when compared to those obtained by the unretargeted CDV-H/F OL pair.

Based on this set of experiments, we selected the hyperfusogenic CDV-H 5804/F SPA pair for further investigation and modification.

## The strength of the CDV H/F interaction inversely correlates with the cell-to-cell fusion efficiency

We postulated that the enhanced cell membrane fusion observed for the CDV-H 5804/F SPA pair might be related to a lower binding avidity at the H/F interface. This was based on the observation that H/F dissociation is essential for the fusion process [23,24]. In order to test our hypothesis, we evaluated the relative strengths of association for different combinations of CDV-H and F proteins by coimmunoprecipitation (co-IP) assays. To facilitate detection, CDV-F SPA was fused to a FLAG-tag [which had no effect on the bioactivity of the protein (S3 Fig, S1 Data)]. The results, presented in Fig 2C, showed that the presence of the M437L amino acid change in CDV-H SPA weakened its affinity for CDV-F SPA. Taken together, these data indicate that there is an inverse correlation between fusogenicity levels and the strength of the CDV-H/F interaction.

## Fully-retargeted CDV envelope glycoproteins display comparable fusion activity to MeV glycoproteins

The CDV-H proteins described above could still use nectin-4 as a receptor (Fig 2A). In order maximize retargeting efficiency, the CDV-H protein needed to be detargeted from this undesirable interaction with human cells. To determine whether the ablation of this natural tropism would affect the cell fusion induced by CDV-H/F binding to a nonnatural receptor, we introduced a Y539A amino acid change to CDV-H, which corresponds to Y543A in MeV-H, an amino acid substitution that was previously shown to abrogate nectin-4-dependent fusion [25] while not affecting cell-surface expression [26]. We then compared the fusion proficiency of CDV-H 5804 (Y539A) / F SPA with that of a fully retargeted MeV-H/F pair [27]. For this, we utilized the quantitative and kinetic fusion assay based on a dual-split GFP/luciferase reporter protein. Our data, presented in Fig 2D and S1 Data, indicate that CD38-targeted CDV-H 5804 (Y539A) had no fusion activity in CHO-nectin-4 cells but did induce fusion in CHO-αHIS cells (construct CDV-H 5804 (Y539A)/F SPA and CDV-H 5804 (Y539A)$^{\alpha CD38}$/F SPA) and CHO-CD38 cells (CDV-H 5804 (Y539A)$^{\alpha CD38}$/F SPA). Because the fusion activity of the nectin-4 blind CDV-H 5804/ F SPA pair was comparable to that obtained with the fully retargeted MeV-H protein, we concluded that CDV-H 5804 (Y539A) can efficiently retarget the CDV-H/F complex to specific receptors, and selected this protein for incorporation in the fully retargeted virus.

## Binding affinity determines efficient retargeting of CDV H/F complexes to CD46

Given that the CDV-H protein selected as described above could be efficiently retargeted to CD38 by fusing a CD38-specific scFv, we next sought to retarget this protein to CD46 via display of a CD46-specific scFv. We postulated that the C-terminal display on CDV-H of an scFv that recognized CD46 with sufficiently high binding affinity would result in CD46-mediated

cell-to-cell fusion activity similar to that induced by the MeV-H/F complex. To test this, we first sought to identify an anti-CD46 scFv with high affinity for CD46 by assessing binding to purified CD46 of several different scFv variants isolated from a phage antibody display library (S4 Fig, S1 Data). We applied surface plasmon resonance technology using sensor chips with covalently immobilized anti-Fc antibody. Chimeric Fc-scFv fusion proteins were captured onto the sensor surface which was subsequently interrogated with soluble CD46 comprising SCR1-4 (Fig 3A, S1 Data). Under these assay conditions, our results showed that the affinity constants ($K_d$) of the chimeric Fc-scFv fusion proteins displaying the A09 and K2 fragments were significantly stronger than those of the K01 and N1E fragments (A09>K2>N1E>K1), primarily due to increased association (A09) or lower dissociation rates (K2) (Table 1, Fig 3B, S1 Data).

We next sought to determine whether fusion of a scFv to the CDV-H protein can support CD46-dependent fusion and if so, how CD46 binding affinity would affect cell fusion. Our primary approach was to perform quantitative fusion assays for the detargeted CDV-H [5804 (Y539)] and retargeted CDV-H [5804 (Y539)-scFv] /F SPA pairs and to compare them to the unmodified MeV-H/F complex on CHO cells and CHO cells expressing either nectin-4 or CD46. All the proteins were expressed at comparable levels (S5A Fig, S1 Data). With the exception of scFv K1, all the other anti-CD46 scFvs allowed the CDV-H/F complex to induce cell-to-cell fusion in CHO-CD46 cells (Fig 3C, S5B Fig, S1 Data) and in a HeLa cell line with high CD46 expression (S6 Fig). As expected, only the MeV-H/F complex induced cell-to-cell fusion in CHO-nectin-4 cells.

We conclude from this set of experiments that there exists a binding affinity threshold for CD46-mediated cell-cell fusion through the retargeted CDV-H/F complexes and above this threshold there is a positive correlation between binding affinity and intercellular fusion.

## The CD46-targeted CDV envelope glycoproteins are efficiently incorporated into MeV virions and mediate virus entry in accordance with their binding affinity

We next investigated whether higher receptor affinity translated into higher virus infectivity. To begin to address this question, we generated a panel of isogenic MeVs where the MeV coat was replaced with CDV-F SPA together with CDV-H 5804 (Y539A) displaying low (K1), intermediate (N1E) and high (A09) affinity scFv specific for CD46 (Fig 4A). These "Stealth" viruses were further engineered to express either eGFP or firefly luciferase as reporter genes and were rescued on Vero-αHIS cells. We next assessed the ability of the Stealth viruses to infect CHO-CD46 cells and either the parental (Vero) or the producer cell line (Vero-αHIS). The data, presented in Fig 4B, showed that all three Stealth viruses were unable to infect the parental Vero cells but they efficiently induced syncytia formation on Vero-αHIS cells, indicating efficient virus replication through the HIS-pseudoreceptor and a lack of interaction with CD46 from African green monkey. On CHO-CD46 cells, only MeV-Stealth-A09 formed a multi-nucleated syncytia pattern. Stealth-N1E gave rise to small clusters of unfused GFP-positive CHO-CD46 cells whereas Stealth-K1 apparently failed to infect them (Fig 4B). When using luciferase as a reporter for infection, we were able to detect CD46-dependent infection with Stealth-K1 but the luciferase levels were significantly lower than those obtained when cells were infected with Stealth-A09 (Fig 4C, S7 Fig, S1 Data). We also observed an affinity-mediated CD46-dependent virus entry in immortalized cell lines routinely used in the laboratory. Again, virus infection in cells expressing human CD46 was observed with Stealth-A09 whereas that of an isogenic and untargeted MeV-Stealth (no scFv) or Stealth-K1/N1E was negligible (S8 Fig, S1 Data). In B95m cells, a marmoset B-cell line widely used for the isolation of

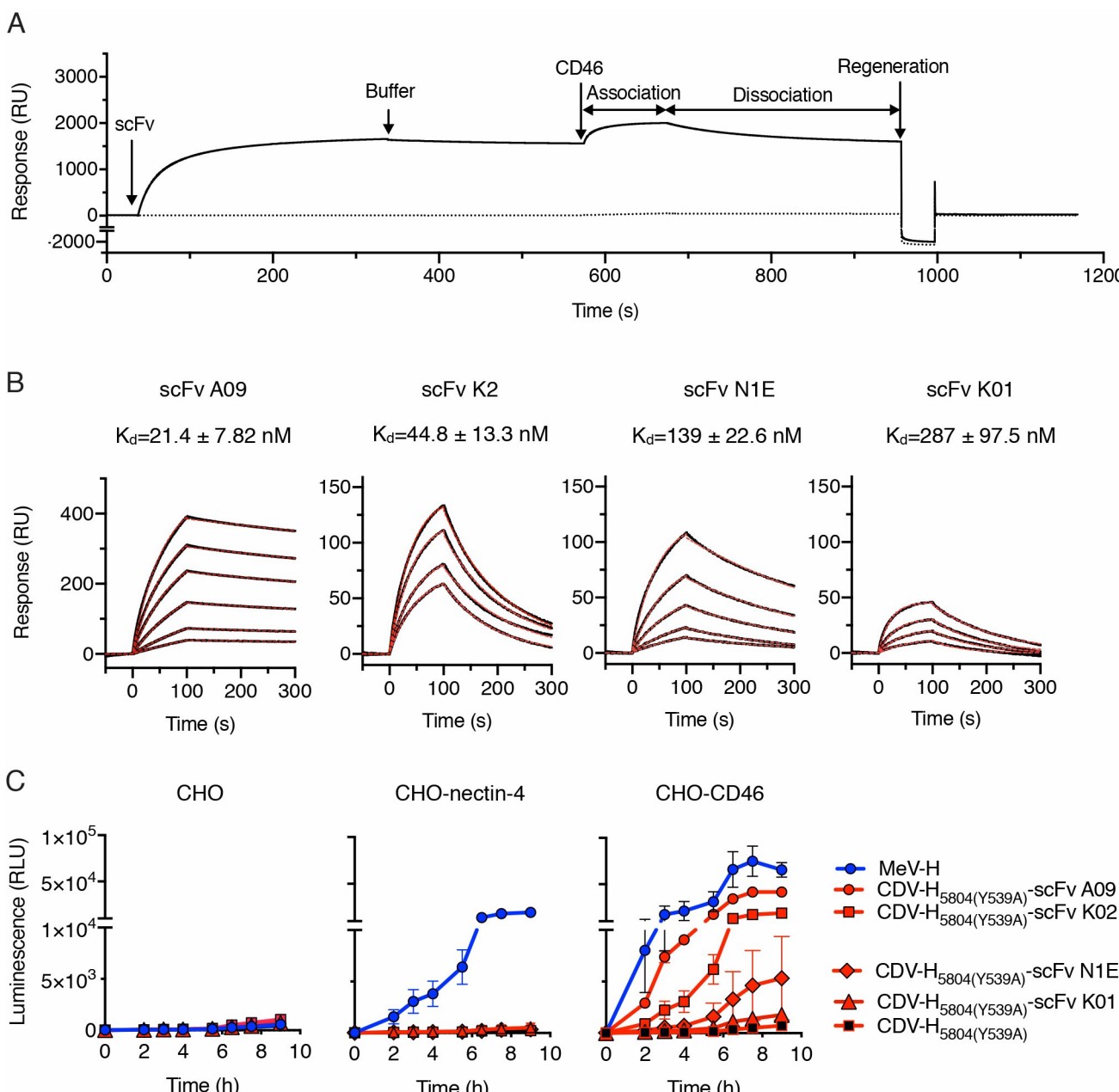

**Fig 3. CD46 binding affinity of the displayed scFv determines CD46-dependent cell-to-cell fusion of the retargeted CDV H/F complex. (A)** Representative sensogram (in resonance units, RU) for the binding of CD46 to biosensor surfaces containing (continuous line) or lacking (discontinuous line) single-chain antibody fragments (scFv). Experimental data represents the injection for 300s of scFv K2 followed by buffer injection. 1 μM of CD46 was subsequently flown over both biosensor surfaces and the signal was recorded during (association) and after injection (dissociation). The surfaces were finally regenerated at the end of the cycle as described in Materials and Methods. **(B)** Binding of CD46 to scFvs as assessed by surface plasmon resonance. Sensograms showing the response units (black lines) of various concentrations of CD46 to the scFvs. Best fit 1:1 binding model is shown as discontinuous red lines. Binding affinity ($K_d$) was determined from the association and dissociation rates (Table 1). **(C)** Quantitative fusion assay for the MeV-H and CDV-H variants on CHO cells. The experiment was carried out in duplicate and repeated twice with similar results (see S5 Fig). The data are shown as the mean ± SD.

measles virus isolates, the lack of the SCR1 domain of CD46 greatly reduced Stealth-A09 infection. On the other hand, infection of B95m cells was readily observed with the unmodified/parental MeV (S8 Fig, S1 Data).

**Table 1. Affinity and kinetic rate constants for single-chain variable fragment binding to CD46.**

| Interaction | $K_{on} \times 10^4$ ($M^{-1}s^{-1}$) | $K_{off} \times 10^{-3}$ ($s^{-1}$) | $K_D$ (nM) |
|---|---|---|---|
| scFv A09-CD46 | $3.34 \pm 1.68$ | $0.606 \pm 0.07$ | $21.4 \pm 7.82$ |
| scFv K2-CD46 | $24.0 \pm 3.79$ | $10.8 \pm 1.81$ | $44.8 \pm 13.3$ |
| scFv N1E-CD46 | $3.69 \pm 1.28$ | $5.16 \pm 2.21$ | $139 \pm 22.6$ |
| scFv K1-CD46 | $4.13 \pm 0.77$ | $11.5 \pm 2.58$ | $287 \pm 97.5$ |

Association ($K_{on}$) and dissociation ($K_{off}$) reaction were determined with a BIACORE T100 instrument using a 1:1 Langmuir binding model.

Based on its superior CD46-dependent virus entry, we chose Stealth-A09 for further characterization. The virus was further propagated to produce virus stock and subjected to Sanger sequencing and protein composition analysis to confirm its identity (Fig 4D). Comparative growth kinetics showed, as expected, that Stealth-A09 replicated in Vero-αHIS cells but not in the parental Vero cell line (Fig 4E, S1 Data). We next fully examined the full receptor-specific

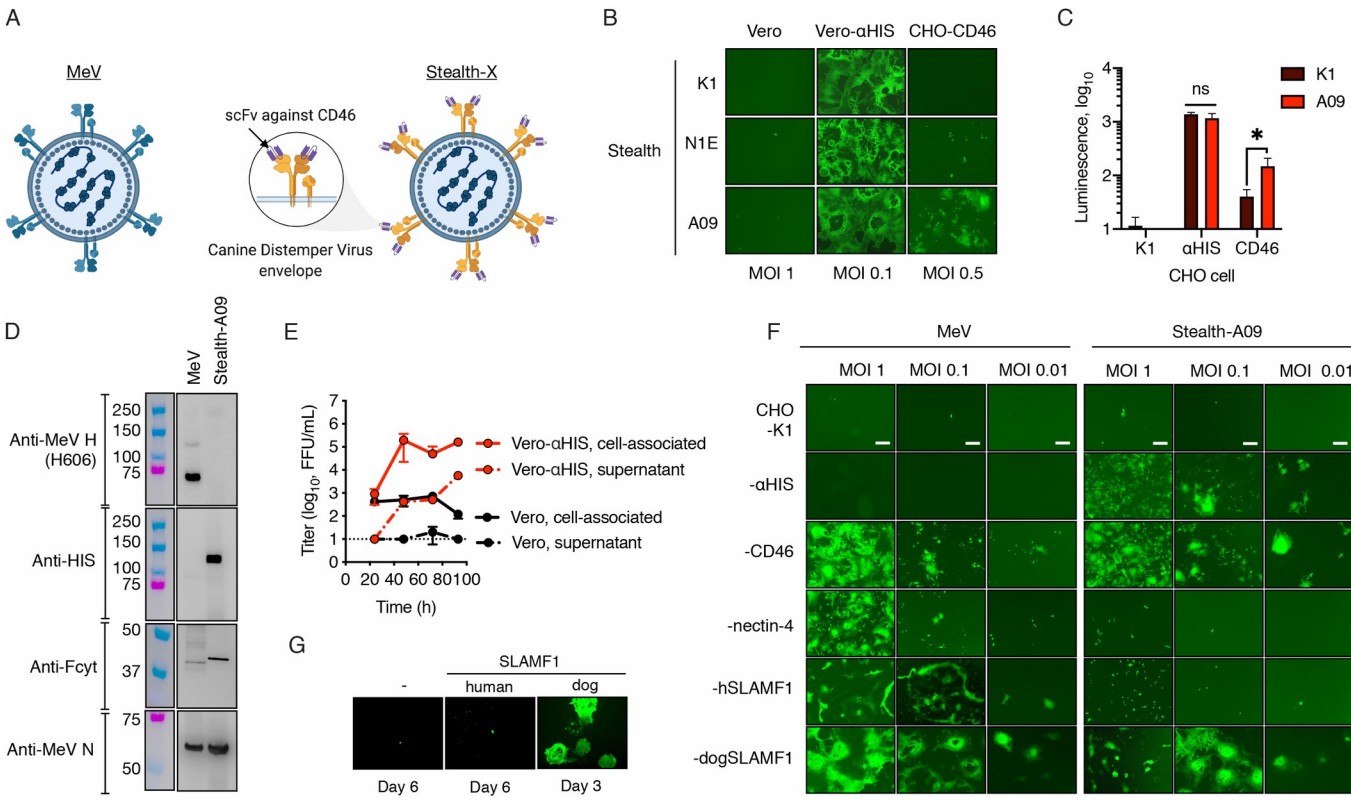

**Fig 4. CD46-retargeted CDV envelope glycoproteins determine virus tropism.** (**A**) Schematic representation of Stealth: a vaccine-derived measles virus pseudotyped with CD46-retargeted CDV H/F envelope proteins. Created with BioRender.com. (**B**) Role of the CD46 binding affinity into virus entry. Cells were infected at the indicated MOI with Stealth viruses displaying scFv with different affinity to CD46. eGFP expression was monitored 48 h post infection. (**C**) CHO cells and derivates expressing the HIS-pseudoreceptor or CD46 were infected with Fluc-expressing Stealth viruses (K1 and A09) at MOI 0.5. Luciferase expression was measured 48 h postinfection. n = 2 except for CHO-CD46 (n = 3), *, p-value<0.05 (two-tailed t-test). (**D**) Protein composition of Vero cells lysates. Western blot analysis was determined with similar amounts of virus particles and probed with the relevant antibodies. The molecular weight of the standard (colored lanes) is indicated. (**E**) Multistep growth kinetics of Stealth-A09 in Vero or Vero-αHIS cells. At the indicated time-point, both the supernatant and the cell pellet were collected and virus titers were determined on Vero-αHIS cells. Values and error bars (SD) were determined for a representative experiment performed in triplicate. (**F**) Virus tropism. CHO cell derivatives were infected with the eGFP-expressing viruses as indicated. eGFP autofluorescence was determined 48 h later. Scale bar, 200 μm. (**G**) Genetic stability of Stealth. Vero-hSLAMF1 cells were infected with Stealth and passaged multiple times. After 8 passages, the recovered virus was used to infect Vero cells expressing human or dog SLAMF1. Representative microphotographs are shown after infection for three (Vero dog SLAMF1) or six days (Vero and Vero human SLAMF1).

tropism of the viruses by infecting a panel of CHO cells stably expressing αHIS, CD46, nectin-4 or either human or dog SLAMF1. The results, presented in Fig 4F, showed that MeV infected CHO cells expressing the receptors CD46 and nectin-4 and either dog or human SLAMF1, whereas Stealth-A09 only infected CHO-αHIS, CHO-CD46 and CHO-dogSLAMF1 cells. Therefore, human CD46 promotes entry and infection by Stealth-A09 in relevant cell lines.

In keeping with a recent report showing that CDV-H does not bind human SLAMF1 [28], we did not observe entry of MeV-Stealth-A09 into CHO-hSLAMF1 (Fig 4F). However, adaptation to human SLAMF1 via a different CDV strain has been observed [29]. Therefore, to assess the likelihood of MeV-Stealth adapting to human SLAMF1, we passaged the virus consecutively in Vero-hSLAMF1 cells and analyzed viral tropism. The infected cells were blind-passaged eight times at 5-day intervals before testing the tropism. This number of passages has been previously shown to suffice to induce cyclical adaptation of measles virus quasispecies [30]. As shown in Fig 4G, at the end of this selective pressure, MeV-Stealth-A09 was still able to induce syncytia formation only in Vero-dogSLAMF1 cells, not in Vero or Vero-hSLAMF1 cells, where only discrete GPF-positive cells were observed. Thus, our data argue against a potential adaptation to allow use of the pathogenic human SLAMF1 receptor.

Collectively, these results indicate that the MeV-H/F glycoproteins can be exchanged with the CD46-retargeted CDV-H/F glycoproteins and that cell entry is dependent on receptor affinity.

## MeV-Stealth oncolytic activity is dependent on its CD46 binding affinity

We next proceeded to determine the antitumor potential of Stealth viruses and the role of CD46 binding affinity *in vivo*. For this, athymic mice bearing peritoneally disseminated SKO-V3ip.1 tumors expressing the firefly luciferase gene (SKOV3ip.Fluc) were treated with a single intraperitoneal dose of saline or $10^6$ TCID50 of Stealth-N1E and Stealth-A09 (n = 5). Tumor burden was then monitored using in vivo bioluminescence imaging. By day 7, we observed a comparable reduction in tumor burden in animals that received either Stealth-N1E or Stealth-A09 when compared with the control group (Fig 5, S1 Data). However, the statistical significance in the reduction of tumor burden was lost for the Stealth-N1E group by day 21, whilst this was not the case for the Stealth-A09 group. Comparison of survival curves showed that only MeV Stealth-A09 increased mouse survival compared to the control group. Thus, Stealth-A09 shows oncolytic potency that is dependent on its higher CD46 affinity.

## MeV-Stealth achieves oncolysis and prolongs survival of myeloma and ovarian tumor-bearing mice

We then assessed the utility of MeV-Stealth-A09 over MeV as an oncolytic agent. For this, we initially left untreated (PBS-treated group) or treated severe combined immunodeficiency (SCID) mice bearing subcutaneous human myeloma xenografts (derived from U266.B1 cells) with a suboptimal intravenous dose of MeV-Stealth or MeV. The tumors in the PBS-treated group continued to grow exponentially, and all mice had to be sacrificed because of the tumor burden by day 12 (Fig 6A, S1 Data). Treatment with either MeV or MeV-Stealth-A09 slowed tumor progression, resulting in a significantly increased median survival time of 7 and 5 days, respectively (Fig 6B, S1 Data). In order to assess whether the oncolytic effect was due to virus replication, we performed histological analysis of explanted tumors. The results, presented in Fig 6C, strongly indicate that both viruses were able to home to tumor tissue. These results strongly suggest that CD46-targeted Stealth induces oncolysis at similar levels compared to MeV, which targets both CD46 and SLAMF1, thereby inducing tumor regression in this multiple myeloma model.

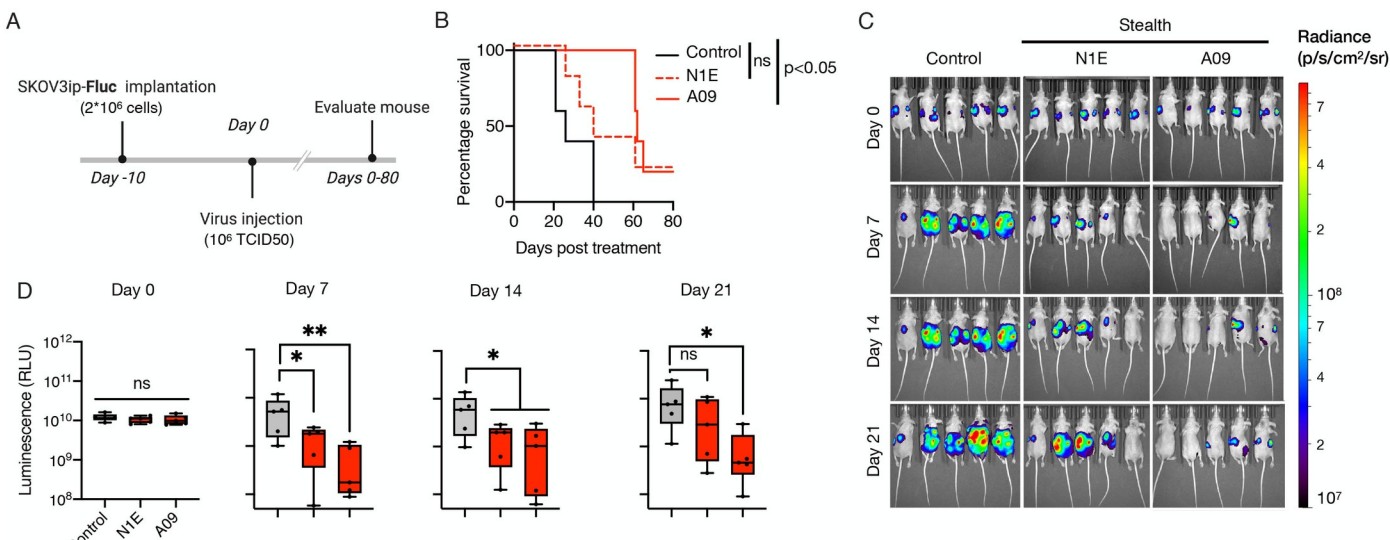

**Fig 5. High CD46 binding affinity governs oncolytic activity of CD46-targeted Stealth virus in a mouse model of ovarian cancer. (A)** Study schematic of the experimental design. SKVOv3ip.1 tumor cells encoding the firefly luciferase gene (SKOV3ip.Fluc) were implanted intraperitoneally into athymic mice. At day 10, $1 \times 10^6$ TCID50 particles of Stealth were administered following the same route. Tumor burden was then monitored at 7 days interval through bioluminescence imaging (BLI). **(B)** Kaplan-survival curves of SKOV3ip.Fluc-bearing mice treated with Stealth-N1E and Stealth-A09 viruses (n = 5). Statistical significance defined by long-rank test. **(C)** Representative BLI showing the dorsal view of treated animals. Radiance (photons per second per cm per steradian, $p/s/cm^2/sr$) was translated to colors to indicate tumor burden in the mice, according to the legend shown on the right. **(D)** Quantification of the total body luminescence in photons per second per cm per steradian ($p/s/cm^2/sr$). n = 5. Ns, not significant; *, p-value $<0.05$; **, $p<0.005$.

Next, we evaluated the therapeutic effect of MeV-Stealth-A09 in prolonging survival in the presence of measles-immune serum. Before embarking on this *in vivo* study, we first evaluated *in vitro* the neutralization sensitivity of recombinant viruses. Our results, presented in S9 Fig and S1 Data, showed that MeV-Stealth-A09 was insensitive to the neutralizing activity of measles-immune human serum, whereas it was completely neutralized by CDV-immune ferret or mouse serum. We observed essentially the opposite pattern for MeV (S9 Fig, S1 Data). We then implanted SKOV3ip.Fluc cells into the peritoneal cavities of athymic nude mice, which received either PBS or measles-immune human serum prior to a single intraperitoneal injection of MeV or MeV-Stealth (Fig 7A). Animals that did not receive any virus injection as well as those treated with MeV in the presence of immune serum exhibited high bioluminescence activity that continued to increase over time (Fig 7C and 7D, S1 Data), indicating that MeV

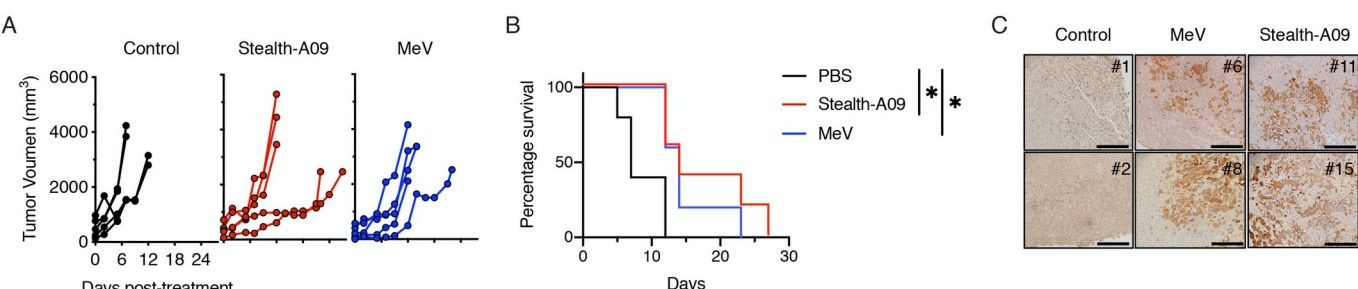

**Fig 6. Stealth-A09 virus achieves oncolysis indistinguishible from the parental MeV in a mouse model of multiple myeloma. (A)** SCID mice bearing subcutaneous U266.B1 cell tumors were treated intravenously with a suboptimal dose of virus. Tumor growth was measured with a caliper (n = 5) and animals were euthanized when the tumors ulcerated or when the tumor size reached 20% of the body weight. **(B)** Kaplan-Meier survival curves (n = 5). Significant differences among the groups were determined by the long-rank test (*, p<0.05). **(C)** Virus trafficking to subcutaneous tumor cells after systemic administration. eGFP expression was evaluated by immunohistochemistry of two representative samples from each group collected at euthanasia. Scale, 200 nm.

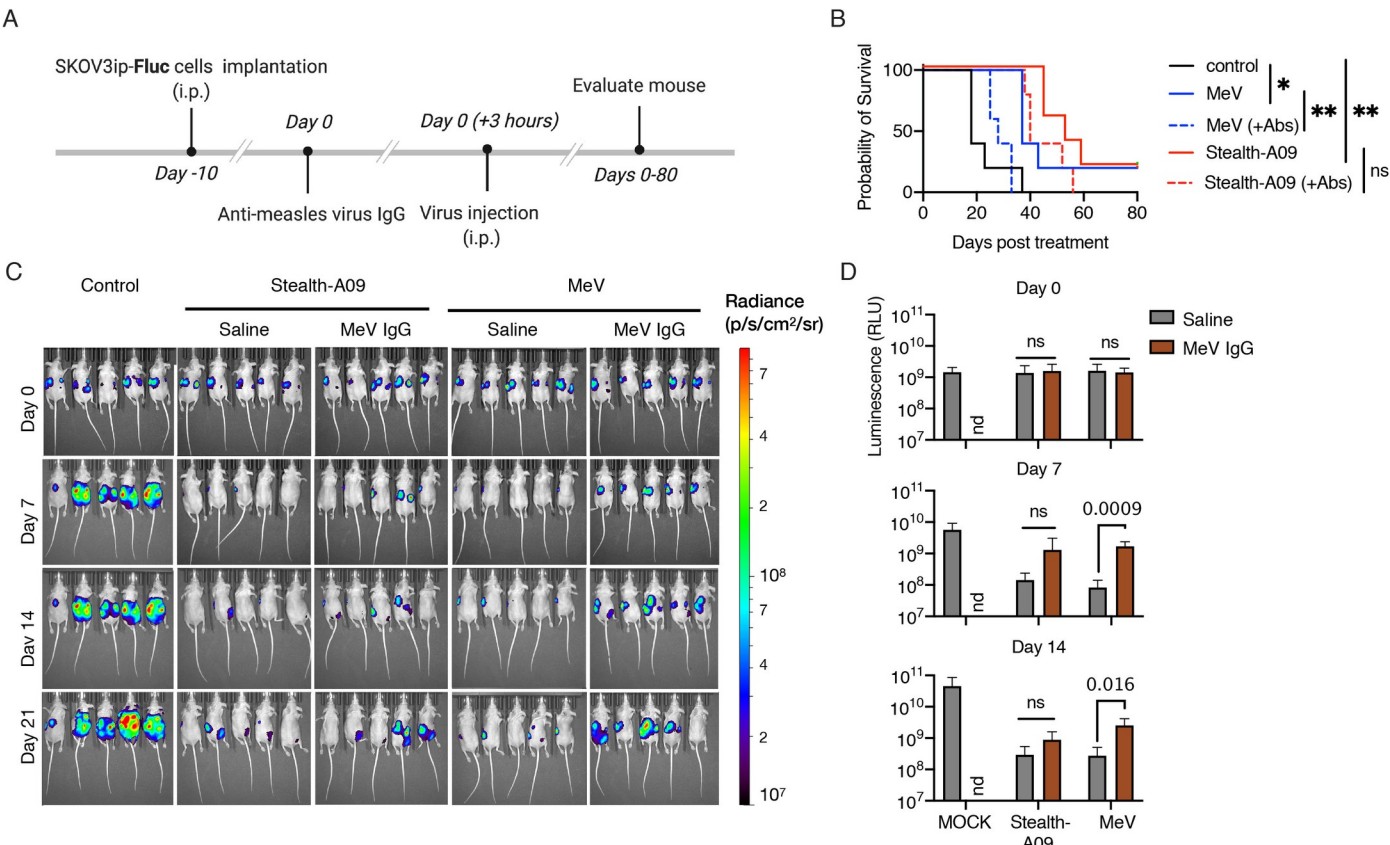

**Fig 7. Stealth virus remains oncolytic in the presence of MeV-immune serum. (A)** SKOV3ip.Fluc cells were injected into athymic nude mice and allowed to establish for 10 days. Next, mice in the relevant groups received 600 mIU of anti-MeV IgG antibody intraperitoneally three hours before virus treatment via the same route. **(B)** Kaplan-Meier survival curves (n = 5 mice per group). Significant differences among the groups were determined by the long-rank test (ns, not significant; *, p<0.05; **, p<0.002). **(C)** Representative BLI showing the dorsal view of treated animals. Radiance (photons per second per cm per steradian, p/s/cm²/sr) was translated to colors to indicate tumor burden in the mice, according to the legend shown on the right. **(D)** Quantification of the total body luminescence in photons per second per cm per steradian (p/s/cm²/sr). n = 5. Statistical significance was determined by one-way ANOVA with Dunnetts' multiple comparison test. Ns, no significant; nd, not done.

could not exert oncolysis in the presence of pre-existent immunity. On the contrary, Kaplan-Meier survival curves indicated that, in the absence of immune serum, both OV treatments significantly prolonged mouse survival, with a median survival times of 37 days for the MeV-treated mice and 53 days for the MeV-Stealth-treated mice versus a median survival time of 18 days for the control mice (Fig 7B, S1 Data). However, prolonged survival following MeV treatment was completely abrogated when mice received measles-immune serum, whereas MeV-Stealth treatment was still able to significantly prolong survival (median survival time of 28 days) with no statistical difference over treatment in the absence of immune serum.

We conclude from this set of experiments that CD46 targeting drives oncolyis in both a xenograft myeloma model and an orthotropic model of ovarian cancer, and that exchange of the MeV coat by the homologous CDV-H/F fusion apparatus shields MeV from MeV-immune human serum.

## Discussion

We here generated MeV-Stealth: a MeV Moraten vaccine strain pseudotyped with CD46-re-targeted CDV that can achieve oncolysis in the presence of human measles-immune serum.

Our findings are consistent with the hypothesis that CD46 tropism governs the efficacy of MeV virotherapy [6,31,32]. The new membrane fusion apparatus comprised of heterotypic CDV-H/F pairs supports antibody-targeted cell fusion at levels similar to those induced by the MeV fusion complex. The CDV envelope was fully retargeted towards human CD46, and did not show any potential to adapt to use of the hSLAMF1 receptor. Finally, MeV-Stealth was oncolytic, its *in vivo* potency was not impacted by measles-immune serum. Collectively, this experimental evidence strongly supports the overall concept that antibody-mediated CD46 targeting of MeV through wild-type CDV envelope proteins is an effective way to circumvent neutralization of oncolytic MeVs.

One of the major obstacles to the production of recombinant MeV carrying foreign glycoproteins is related to the necessity of proper trafficking and assembly of all the viral components [20,33]. We and others have previously reported that the envelope of the CDV OL strain is fully interchangeable with that of MeV [11,14,16]. However, Rouxel et al.,[13] succeeded in creating a chimeric MeV strain with the wild-type CDV envelope by replacing not only the MeV-H/F pair but also MeV-M. Thus, it is possible that the heterologous MeV-M protein that we used in this study drives differential particle formation through a relatively weak interaction with the CDV H/F cytoplasmic tail. However, our western blot analysis did not seem to indicate an abnormal particle-to-PFU ratio, suggesting that the heterotypic CDV-H/F envelope glycoproteins are efficiently incorporated into the virus particles.

Since the fusogenic activity of envelope glycoproteins is another key determinant for the generation of replication-competent MeV [34], we sought to enhance the fusion activity of the wild-type CDV envelope to levels similar to those of the OL envelope. We observed that homotypic H/F pairs from two different CDV strains (the Arctic and Europe/South America-1 genetic groups) exhibited comparable but distinct heterologous fusion phenotypes compared with the phenotype of the OL strain. Until now, similar results have been documented only across the genus *Henipavirus*, including Nipah virus, Hendra virus and Ghana virus [24,35,36]. Our results using a fully retargeted Nipah-G/F glycoprotein complex [21] are in agreement with the reduced fusogenicity of this complex compared with that of the CDV-H/F complex [36,37]. However, no previous studies accounted for the different cognate receptors of Nipah virus and CDV on the cell lines used [36,37].

The overexpression of CD46 in many different human cancers makes it a reasonable target of OVs and antibody-drug conjugated therapies [38–40]. We have previously established that cell lysis induced by MeV glycoproteins occurred only above a certain CD46 surface expression threshold, confirming the tumor selectivity of the virus [6,31]. In the present study, cell fusion was observed only when we added an anti-CD46 scFv to CDV-H with a binding affinity ~ 100nM, which compares to that reported for the MeV-H/CD46 interaction [41,42] and suggests an affinity threshold for cell fusion, as we have previously reported for HER2/neu [43]. Thus, our findings establish the basis for increasing tumor selectivity through the use of affinity-tuned scFvs. Indeed, toxicities observed with chimeric antigen receptor (CAR) T cell therapies have been ameliorated through the use of low-affinity scFvs against targets [44,45]. Since CD46 is expressed by all nucleated human cells, the use of low-affinity CD46-specific scFvs might also alleviate safety concerns when moving into clinical trials.

In contrast to wild-type CDV strains, MeV-Stealth appears to be relatively resistant to adaptation to the use of alternate cell surface receptors. Pseudotyping approaches using foreign envelopes from viruses with unknown tropism limit the transition from bench to bedside due to the difficulty of studying the zoonotic potential of the pseudotyped products [16,20]; therefore, envelopes with known tropism, whether in humans or animals, are preferred [21,46,47]. CDV is a promiscuous morbillivirus, and adaptation to the use of monkey and human SLAMF1 has been documented [29,48]. The lack of adaptation to allow efficient use of human

SLAMF1 by MeV-Stealth might therefore be related to the increased genetic stability of MeV compared with that of CDV [49]. Indeed, while MeV genetic diversity has decreased over time, new genetic lineages of CDV are frequently discovered [50,51].

Finally, the envelopes developed here have implications for improving gene and cell therapies. Our group previously showed that vesicular stomatis virus (VSV) pseudotyped with the MeV envelope has superior antitumor efficacy over either MeV or VSV [52,53]. Similarly, arming adenovirus and herpes simplex virus strains with the MeV envelope glycoproteins enhances their cytotoxicity to tumor xenografts, likely via cell fusion-induced immunogenic cell death [22,54–56]. We expect that the new envelope generated here will be easily applicable to the aforementioned virus vectors to advance their *in vivo* applicability in MeV-immune individuals.

One of the weaknesses of this study was our inability to examine the immunotherapeutic phase of MeV virotherapy due to the lack of reliable immunocompetent models. Mouse cells do not allow MeV replication [57], and our observations in SCID and athymic nude mice accounted for only direct viral oncolysis. However, based on the existing literature on Newcastle Disease virus [58], reovirus [59], herpes simplex virus [60], and Maraba virus [61], we anticipate a synergistic effect of MeV-Stealth treatment in MeV-immunized mice due to cross-reactive T-cell responses between MeV and CDV [14,62,63]. A second weakness of the study relates to the mechanism of antibody mediated protection against measles. Here we focused exclusively on direct virus neutralization due to the good correlation between antibody titers, as determined by virus neutralization assay, and protection from measles [64–66]. Thus, we did not evaluate alternative mechanisms of antibody mediated protection such as antibody-dependent complement-mediated cytolysis and antibody-dependent cellular-mediated cytotoxicity (ADCC)[67–69].

In summary, the data presented in this manuscript indicate that substitution of the MeV envelope glycoproteins by CD46-targeted CDV envelope glycoproteins is an alternative strategy to circumvent the inhibition of MeV-immune serum and achieve oncolysis in multiple myeloma and ovarian cancer models. Future use of CDV envelopes displaying affinity-tuned CD46 binding domains might facilitate the development of Stealth platforms for clinical studies by sparing noncancerous human cells with basal levels of CD46.

## Materials and methods

All experiments were carried out in accordance with US regulations and approved by the Mayo Clinic Institutional Animal Care and Biosafety Committee (IACUC).

### Cell lines

H1-HeLa (Cat. # CRL-1958, ATCC, Manassas, VA, USA), A549 (Cat. # CCL-18, ATCC), BHK (Cat. # CCL-10, ATCC), HEK293T (a kind gift from Dr. François-Loïc Cosset, Université de Lyon), SKOV3ip.1 and SKOV3ip.1-Fluc [70] were maintained in Dulbecco's modified Eagle's medium (DMEM; Cat. # SH30022.01, GE Healthcare Life, Pittsburg, PA, USA) supplemented with 10% fetal bovine serum (FBS; Cat. #10437–028; Thermo Fisher Scientific, Waltham, MA, USA). HT-29 cells obtained from Dr. Donald Kohn (University of California) were grown in McCoy's 5A medium (Cat. # 30–2007, ATCC), 10% FBS. Vero African green monkey kidney cells (Vero, Cat. # CCL-81, ATCC) and their derivatives (expressing human nectin-4 [71], human SLAMF1 [72] or a membrane-anchored single-chain variable fragment (scFv) specific for a hexahistidine peptide (6× HIS-tag) [27] were cultured in DMEM as previously described [14]. Vero cells constitutively expressing the dog SLAMF1 molecule (Vero-dogSLAMF1) were generated by transduction and puromycin selection of a second-generation lentiviral vector

(kindly provided by Dr. Lukkana Suksanpaisan [Imanis Life Science, Rochester, MN, USA]) encoding, under the control of the spleen focus-forming virus promoter, a codon-optimized SLAMF1 molecule from *Canis lupus familiaris* (GenBank NP_001003084.1) with an N-terminal FLAG-tag sequence (DYKDDDD). Cells were maintained in DMEM supplemented with 5% FBS. The Chinese hamster ovary (CHO) cell line, CHO-CD46 cells [22], CHO-hSLAMF1 cells [73], CHO-dogSLAMF1 cells [74], CHO-nectin-4 cells [75], CHO-αHIS cells [27], CHO-CD38 cells [76], U266.B1 (kindly provided to us by Dr. David Dingli [Mayo Clinic, Rochester, MN]), HT2009 (a kind gift from Dr. Manish Patel [University of Minnesota] and the CD46-SCR1 domain deficient marmoset B lymphoblastoid cell line B95m [77,78] were grown in RPMI 1640 medium supplemented with 10% FBS as described elsewhere. Cells were incubated at 37˚C in 5% CO2 with saturating humidity.

## Plasmids and construction of full-genome recombinant measles virus (MeV)

To generate canine distemper virus (CDV) SPA.Madrid/22458/16 expression plasmids, total RNA was extracted from CDV SPA.Madrid/22458/16 isolate–infected Vero/dog SLAMF1 cells (passage 1) using the RNeasy Mini Kit (Qiagen, Hilden, Germany). Both the CDV-Hemagglutinin (H) and CDV-Fusion (F) genes were reverse transcribed with SuperScript III Reverse Transcriptase (Cat. # 11752050, Thermo Fisher Scientific) and amplified by PCR with the following primers: CDVH7050(+): 5'-AGAAAACTTAGGGCTCAGGTAGTCC;-3' CDVH8949 (-): 5'-TCGTCTGTAAGGGATTTCTCACC-3'; CDVF4857(+): 5'-AGGACATAGCAAGCCAACAGG-3' and CDVH7050(-): 5'-GGACTACCTGAGCCCTAAGTTTTCT-3'. PCR products were sequenced directly by Sanger sequencing (Genewiz, Plainfield NJ, USA) and cloned into the pJET1.2 vector (Thermo Fisher Scientific). Next, the CDV-H open reading frame was PCR amplified with a forward primer (5'-CCG GTA G<u>TT AAT TAA</u> AA*C TTA GGG TGC AAG ATC ATC GAT A*AT GCT CTC CTA CCA AGA TAA GGT G-3') and a reverse primer (5'-CTA TTT CAC <u>ACT AGT</u> *GGG TAT GCC TGA TGT CTG GGT GAC ATC ATG TGA TTG GTT CAC TAG CAG CC*T CAA GGT TTT GAA CGG TTA CAG GAG-3') and cloned into a PacI and SpeI-restricted (New England Biolabs, Ipswich, MA, USA) pCG vector [79] using an InFusion HD kit (Takara, Shinagawa, Tokyo, Japan). The primers contained the PacI and SpeI restriction sites (underlined) as well as the coding sequence for the untranslated region of MeV-H (italics). Similarly, the CDV-F open reading frame (amino acid residues 136–662) was cloned into the HpaI/SpeI-restricted pCG-CDV-F plasmid [11]. The resulting plasmid pCG-CDV-F SPA.Madrid/22458/16 contained coding sequences for the MeV-F untranslated region and signal peptide.

Expression plasmids for the CDV-H/F Onderstepoort vaccine and 5804P isolate [11], as well as MeV Nse strain, are described elsewhere [79]. The signal peptide for CDV-F 5804 was replaced with heterologous MeV-F as described above for CDV-F SPA.Madrid/22458/16. The open reading frames of the Nipah-G and Nipah-F glycoprotein genes were amplified from purchased RNA templates (Cat. # NR-37391, BEI Resources), and the Nipah-F gene (GenBank AF212302.2) was inserted into the pCG vector using the NarI and PacI sites. Retargeted versions of the H/G proteins were generated by inserting the homologous PacI/SfiI-digested PCR product into pCGHX α-CD38 [76]. Insertion of the coding sequence for an scFv recognizing CD46 was performed by exchanging the anti-CD38 scFv via the SfiI and NotI restriction sites. Site-directed mutagenesis (QuickChange Site-Directed Mutagenesis Kit, Agilent Technologies, Santa Clara CA, USA) was used to ablate the tropism of H and remove the SpeI site in CDV-F.

The viruses used in this study were derived from the molecular cDNA clone of the Moraten/Schwartz vaccine strain pB(+)MVvac2(ATU)P, with an additional transcriptional unit

downstream of the phosphoprotein gene [79,14]. The plasmid backbone was replaced with the pSMART LCkan vector (Cat. # 40821–1; Lucigen, Middleton, WI, USA), with an optimized T7 promoter followed by a self-cleaving hammerhead ribozyme (Hrbz) [14,80]. eGFP or firefly luciferase were cloned into the infectious clone by using the unique MluI/AatII restriction sites. To generate a cDNA copy of the MeV Moraten carrying Nse-derived glycoprotein, the PacI/SpeI and NarI/PacI fragments from the pCG-H and pCG-F plasmids [79], respectively, were cloned into pSMART-MVvac2(GFP)P. Rescue of rMeVs was carried out employing the STAR system [27].

## Expression of recombinant proteins

A plasmid encoding the CD46-Fc fusion protein was produced by fusing the CD46 ectodomain (residues 35–328) with the Fc domain of IgG1 (Cat. # pfc1-hg1e3; InvivoGen, San Diego, CA, USA). The scFvs K1, K2 and A09 were designed with the VL and VH sequences separated by a GSSGGSSSG flexible linker, codon-optimized, synthesized and cloned into pUC57-Kan (GenScript). A fourth scFv (N1E) was designed with the VH and VL sequences separated by an SSGGGGS linker, codon-optimized, synthesized by Creative Biolabs (Shirley, NY) and cloned into pCDNA3.1+ (Invitrogen). For the IgG constructs, scFvs were cloned into the unique AgeI and KpnI sites of pHL-FcHIS (Cat. # 99846, Addgene, Cambridge, MA, USA), harboring the coding sequence for a secretion signal and a C-terminal human Fc region followed by a 6× HIS-tag. The recombinant proteins were expressed by transfecting Expi293F suspension cells (Thermo Fisher) in serum-free Expi293 expression medium (Thermo Fisher) in shaker flasks following the manufacturer's instructions. The culture supernatants containing the recombinant proteins were collected and passed through a Protein G chromatography cartridge (Cat. # 89926, ThermoFisher). Bound recombinant proteins were eluted with 0.1 M glycine (pH 2.0), followed by immediate neutralization with 1 M Tris (pH 8.0), and the isolated proteins were concentrated with an Amicon Ultra centrifugal concentrator (Millipore Sigma, Burlington, MA, USA). CD46 and nectin-4 were released from the Fc region by incubation with HRV 3C Protease (Thermo Fisher) at a 1:200 ratio. A final purification step was performed using a Superdex 75 10/300 gel filtration column (GE Healthcare) equilibrated in phosphate-buffered saline (PBS). Protein concentrations were calculated from the protein extinction coefficient as determined from the amino acid composition.

## Fusion assays

Cells were transfected using Fugene HD (PROMEGA, Fitchburg WI, USA) or TransIT-LT1 transfection reagent (Mirus Bio LLC, Madison WI, USA). For a quantitative fusion assay, a dual-split reporter system [81,82] was used as described elsewhere [14], using BHK cells as the effector cells. For semiquantitative assessment of fusion, Vero cells and derivative cell lines were transfected with a total of 0.1 µg of DNA (1:1 ratio of H and F expression plasmids), including a GFP expression plasmid for added visualization of syncitia formation. Images were obtained with a microscope (Eclipse Ti-S; Nikon) at 40x or 100x magnification.

## Expression analysis of morbillivirus attachment proteins

For assessment of the level of the H polypeptide, transfected cells were analyzed by flow cytometry or cellular enzyme-linked immunosorbent assay (CELISA) using an anti-6× HIS-tag monoclonal antibody (Cat. # 130-120-787, Miltenyi Biotec or Cat. # MA1-135, Thermo Fisher Scientific), as described elsewhere [14,83]. For analysis of total protein expression by flow cytometry (FACSCanto, BD Biosciences, San Jose, CA, USA), cells were treated with the

eBioscience Intracellular Fixation & Permeabilization buffer (Cat. # 88-8823-88, Thermo Fisher Scientific).

## Envelope glycoprotein coimmunoprecipitation (co-IP)

Three micrograms (1 μg of H and 2 μg of F) of total DNA were transfected into HEK293T cells ($4 \times 10^5$ cells). After 24 h, the cells were washed twice with PBS and treated with the cross-linker 3–3'-diothiobis (sulfosuccinimidyl propionate) (DTSSP; Cat. # 21578, Thermo Fisher Scientific) at 1 mM, followed by quenching with 20 mM Tris/HCl (pH 7.4) and lysis with 0.4 mL of M-PER mammalian protein extraction reagent (Thermo Fisher Scientific) containing a 1× Halt protease and phosphatase inhibitor cocktail (Cat. # 1861281, Thermo Fisher Scientific). Soluble fractions were collected after centrifugation at 10,000 x g for 10 min at 4˚C, and one-thirtieth of the volume was set aside as the cell lysate input. The rest was incubated with 0.5 μg of anti-FLAG monoclonal antibody M2 (Sigma-Aldrich) and EZview red protein G affinity gel (Sigma-Aldrich, St. Louis, MO, USA). The precipitated material was washed (20 mM Tris-HCl, pH 7.4, 140 mM sodium chloride) and denatured by boiling in laemmli buffer containing β-mercaptoethanol.

## SDS-PAGE and immunoblotting

Samples were fractioned by gel electrophoresis on a 4 to 12% NuPAGE Bis-tris gel (Thermo Fisher) and transferred to polyvinylidene difluoride (PVDF) membranes using an iBLOT 2 dry blotting system (Cat. # IB21001, Thermo Fisher Scientific). The protein material was detected though incubation with the antibodies anti-MeV-H606 [20], anti-MeV-F431 [84], anti-Fcyt [11], anti-MeV-N [85], anti-HIS (Cat. # A01857-40, GenScript, Piscataway NJ, USA), anti-β-actin (Cat. # A3854, Sigma-Aldrich), and anti-CD46 (Cat.# sc-7056, Santa Cruz, Dallas TX, USA). Immunoblots were visualized using a rabbit horseradish peroxidase (HRP)-conjugated secondary antibody and KwikQuant Imager (Kindle Bioscience LLC, Greenwich CT, USA). Representative results of two independent repeats are shown.

## Antibody binding assay

An enzyme-linked immunosorbent assay (ELISA) was used to measure the binding of scFvs to CD46. Nunc-Immuno MicroWell 96-well solid plates were coated overnight at 4˚C with 1 μg of purified CD46 or nectin-4 in 0.05 M carbonate-bicarbonate buffer, pH 9.6. (Cat. # E107, Bethyl Laboratories, Montgomery TX, USA). Purified scFv-Fc fusion proteins were then diluted in PBS and added at a concentration of 12.5 μg/mL. Bound antibodies were detected with a secondary anti-human IgG (Fc-specific) HRP-conjugated antibody (1:70,000; Cat. # A0170, Sigma-Aldrich). In parallel, 125 ng of scFv-Fc fusion protein was first bound to wells, and protein levels were monitored by measuring the optical density ($OD_{490\ nm}$) after incubation with the secondary antibody alone.

## Surface plasmon resonance (SPR)

The interaction between the scFv A09, N1E, K1, K2 and CD46 was measured using series S CM5 sensor chips on a Biacore T-100 system (GE Healthcare, Waukesha, WI, USA). For A09, N1E, K1 and K2, 50 μg/mL of an anti-Fc antibody (Cat. # MAB1302, EMD Millipore, Burlington, MA, USA) diluted into 10 mM NaAcetate, pH 4.5, were immobilized to the active and the reference channel of the CM5 chip using amine coupling kit reagents (EDC (1-ethyl-3-[3-dimethylaminopropyl]carbodiimide), NHS (N-hydroxysuccinimide) and ethanolamine). The immobilization of the antibody resulted in ~12,000 response units. The interactions

between CD46 and the anti-Fc antibody captured scFv were measured at 25˚C with a data rate of 10 Hz using HBS EP buffer (0.01 M HEPES pH 7.4, 0.15 M NaCl, 3 mM EDTA, 0.005% v/v Surfactant P20). Each binding cycle began with the loading of 15 µg/mL of the scFv onto the active channel for 300 s at a flow rate of 10 µL/min. After 100 s of buffer wash and a 120 s stabilization period, CD46 (concentration range 50 nM- 1,000 nM for A09, 37.5 nM-100 nM for K2 and 50–1,000 nM for N1E and K1) was flown over the active and the reference channel for 100 s at a flow rate of 40 µL/min. The association phase was followed by a 200 s dissociation period followed by a 60 s injection of 10 mM glycine pH 2.0 at a flow rate of 30 µL/min, to regenerate the surface immobilized anti-FC antibody. All sensograms were fitted with a 1:1 binding model using the Biacore T100 evaluation software v2.04.

## Infections and virus growth kinetic analysis

For virus infections, cells were infected at the indicated multiplicity of infection (MOI) for 90 min at 37˚C in Opti-MEM I reduced-serum medium. After the absorption phase, we removed the inoculum, washed, and added viral growth media (DMEM+5% FBS). When using eGFP-expressing viruses, fluorescence microscopy photographs were taken 48 hours post-infection. For analysis, plates were imaged using a Celígo Imaging Cytometer (200-BFFL-4B, Nexcelom Biosciences). Fluorescence forming units were identified and the number was normalized to the mean obtained in Vero-αHIS. For infections with Fluc-expressing viruses, luciferase expression was measured using an Infinite M200 Pro multimode microplate reader (Tecan Trading AG) after adding 0.5 mM of D-Luciferin to the infected cells.

For virus growth kinetic analysis, Vero cells and derivative cell lines seeded in 6-well plates 16–18 h prior to infection were infected at a MOI of 0.03 for 90 min in Opti-MEM I (Cat. # 31985070, Thermo Fisher Scientific). The inoculum was then removed, and the cell monolayers were washed three times with Dulbecco's phosphate-buffered saline (DPBS; Cat. # MT-21-031-CVRF, Mediatech, Inc., Manassas, VA, USA), and the medium was replaced with 1 mL of DMEM supplemented with 5% FBS. At the indicated time points, cell supernatants were collected, and cells were scraped into 1 mL of Opti-MEM I, followed by 3 freeze/thaw cycles. Cell debris was removed by centrifugation (2,000 x g for 5 min), and virus titers were determined in Vero-αHIS cells.

## Fluorescence-activated cell sorting analysis and quantification of cell surface molecules

Cells were detached by using TrypLE Express enzyme (ThermoFisher) and immediately incubated with phycoerythrin-conjugated antibodies: anti-SLAMF1 (Cat. # FAB1642P; R&D Systems), anti-CD46 (Cat. # FAB2005P; R&D Systems), anti-nectin-4 (Cat. # FAB2659P; R&D Systems), or control isotype antibody (Cat. # IC0041P; R&D Systems). After a 1-hour incubation at 4˚C, cells were washed and fluorescence was measured in a FACSCanto flow cytometry system (BD Bioscience). The number of receptors per cell was estimated with calibration beads (BD QuantiBRITE; BD Biosciences) as the reference standard.

## Immunization studies

Four to six-week-old male and female Ifnar[tm]-CD46Ge mice [86], deficient for type I IFN receptor and transgenically expressing human CD46, were inoculated intraperitoneally (i.p.) with $1 \times 10^5$ TCID50 particles of MeV or Stealth-A09. On day 28, serum samples were collected and stored at -20˚C until assessed for neutralizing antibodies.

## Neutralization assays

A fluorescence-based plaque reduction microneutralization (PRMN) assay was carried out as previously described [14,87]. Briefly, Vero-αHIS cells were seeded in a 96-well plate, and serial dilutions of serum samples were premixed for 1 h at 37°C with virus inoculum before they were added to the cells. The data were plotted as the log (dilution of serum) vs. the normalized response (variable slope) present with GraphPad software (Prism 8) and the neutralization dose 50% was calculated (ND50). Inclusion of the 3rd World Health Organization International serum standard (3IU/mL) enabled conversion of antibody titers to mIU/mL by calculation of the unitage constant [88]. Pooled human serum from 60–80 donors that had blood type AB (Cat. # HS1017; Lot # C80553, Valley Biomedical Inc., Winchester, VA, USA) was used. The following reagents were obtained from the NIH Defense and Emerging Infections Research Resources Repository, NIAID, NIH: polyclonal anti-MeV antibody, Edmonston, (antiserum, Guinea Pig), Cat. # NR-4024 and polyclonal anti-CDV Lederle Avirulent (antiserum, Ferret), Cat. # NR-4025.

## Experimental oncolytic therapy

To establish subcutaneous tumors, 6-week-old female severe combined immunodeficiency (SCID) mice were injected in the right flank with $1 \times 10^7$ U266.B1 tumor cells. When the tumor reached 0.5 cm in diameter, mice received a single intravenous dose of MeV (n = 5) or Stealth (n = 5) at $1 \times 10^5$ 50% tissue culture infectious dose (TCID50). Control mice (n = 5) were injected with an equal volume of PBS. Animals were euthanized when the tumors ulcerated or when the burden reached 20% of the body weight. Tumor diameter was measured every other day, and tumor volume was calculated with the formula length x length x width x 0.5.

To establish an orthotopic model of ovarian cancer, $5 \times 10^6$ SKOV3ip.1 cells expressing firefly luciferase (SKOV3ip.1-Fluc) were injected into the peritoneal cavity of athymic nude mice. Ten days later, the animals received 600 mIU of measles-immune serum (Cat. # HS1017; Lot # C80553, Valley Biomedical Inc.) or an equal volume of saline, and three hours later, they were treated with a single intraperitoneal dose ($1 \times 10^6$ TCID50) of MeV (n = 5) or Stealth (n = 5). The mice in the control group received a similar volume of Vero cell lysates (n = 5). For the therapy experiment, $5 \times 10^6$ SKOV3ip.1-Fluc cells were implanted instead. The tumor burden was monitored weekly through *in vivo* bioluminescence imaging using an IVIS Spectrum instrument (Perking Elmer, Waltham, MA, USA). The mice were euthanized at the end of the study (80 days), when they developed ascites or had lost 20% of their body weight. Statistical comparisons among groups were performed with the log-rank (Mantel-Cox) test, and p<0.05 was considered statistically significant.

## Statistical analysis

Statistical analyses were performed with GraphPad Prism 8.3.1 version for Mac OS X. Significant differences among groups were determined using either a one-way analysis of variance (ANOVA) with Holm-Sidak's multiple comparison test or a two-way ANOVA with Turkey's multiple comparison test, as indicated in the corresponding figures. Survival data were analyzed using the Kaplan-Meier method, and the log-rank test was used to identify significant differences among groups.

## Supporting information

**S1 Fig. Conservation of amino acid residue M437 in the CDV-H protein among different genetic groups.** Sequence alignment was performed with CDV-H sequences retrieved from

GenBank, including the CDV-H sequence determined here for the SPA.Madrid/22458/16 isolate. The accession numbers are indicated.
(TIF)

**S2 Fig. Integrity and expression of chimeric ligand-displaying receptor binding proteins.**
**(A)** Western blot analysis of HEK293T cells transfected with the indicated proteins fused to an anti-CD38 scFv or not. Proteins were blotted with an anti-HIS antibody or an anti-β-actin antibody (loading control). (**B**) Protein expression of the attachment proteins and mutants on HEK293T cells fixed with or without permeabilization analyzed by flow cytometry. Histograms are from one representative experiment out of two biological replicates. Geometric mean intensity ± SD from two biological replicates is shown at the upper right corner of each histogram. Filled curves denote cells transfected with empty plasmids.
(TIF)

**S3 Fig. FLAG tag insertion in the F ectodomain and its effect on protein bioreactivity. (A)** Schematic drawings of uncleaved MeV-F and CDV-F. The $NH_2$ and COOH termini, signal peptide (SP), fusion peptide (FP), and transmembrane (TM) and cytoplasmic regions are indicated. The sequence surrounding the cleavage site (in bold) and that of the fusion peptide are shown. The numbering considers the homotypic signal peptides. **(B)** Syncytia formation in Vero cells after cotransfection of homologous H and F expression plasmids with FLAG insertions at different positions. Cells were stained at 16 hours posttransfection, and microphotographs were acquired for quantification. **(C)** Quantification of syncytia formation. The data are shown as the mean ± SD (n = 20). Significance was determined using one-way ANOVA with Holm-Sidak's multiple comparison test (ns, not significant; ***, p≤0.001). **(D)** Dual-split protein fusion assay for the cotransfection of CDV-H/F SPA with or without a FLAG-tag insertion at aa 216. The luciferase signal was measured at 8 hours. The experiment was performed in technical duplicates.
(TIF)

**S4 Fig. CD46 specificity of scFvs. (A)** SDS-PAGE analysis of target proteins. MW: molecular weight ladder, C: Coomassie staining; WB: western blot analysis using an anti-CD46 antibody. **(B)** Size exclusion chromatography trace for the CD46 used in the experiments. The estimated MW from a calibration curve is indicated**. (C)** Binding of scFv-Fc tagged fusion proteins to CD46 or nectin-4 as determined by ELISA. Detection was performed with the Fc portion used as a control for the amount of protein. Experiments were performed in technical duplicates. The data are shown as the mean ± SD, n = 2). Significance was determined using one-way ANOVA with Holm-Sidak's multiple comparison test. *, p<0.05; **, p<0.005.
(TIF)

**S5 Fig. Related to Fig 3. Binding affinity of the scFv displayed onto the CDV-H/F complex drives enhanced cell-cell fusion. (A)** Cellular enzyme-linked immunosorbent assay (CELISA) for the amount of cellular protein used in the quantitative fusion assay on Fig 3C. A CELISA was performed on CHO cells transfected with the indicated attachment protein using an anti-6× HIS-tag monoclonal antibody (n = 5). **(B)** Quantitative fusion assay for the CD46-retargeted CDV-H/F complex using affinity tuned scFvs (same data as presented in Fig 3C). Y539A indicates the substitution in CDV-H to ablate the natural tropism for nectin-4.
(TIF)

**S6 Fig. Assessment of receptor interactions for the engineered CDV fusion apparatus complex.** Cells were cotransfected with MeV-F and MeV-H or CDV-F and CDV-H retargeted variants with a CD46-specific scFv. For visualization purposes, an expression plasmid encoding

eGFP was cotransfected, and eGFP autofluorescence was visualized at 24 hours posttransfection. Y539A indicates the substitution in CDV-H to ablate the natural tropism for nectin-4. "+" and "–" symbols were used for semiquantification (same as presented in Fig 2A). "Undisplay" indicates no scFv.
(TIF)

**S7 Fig. Related to Fig 4C. Increased binding affinity to CD46 enhances CD46-specific virus entry.** Fluc-expressing Stealth viruses were used to infect the indicated cells at decreasing MOI. Luciferase expression was measured 48h postinfection. n = 2 for all except CHO-CD46 and Stealth-A09 (n = 3).
(TIF)

**S8 Fig. Affinity-mediated, CD46-dependent infection of cell lines. (A)** A panel of immortalized cell lines was infected with parental MeV, MeV-Stealth and MeV-Stealth displaying CD46-specific scFv-A09 (Stealth-A09). Representative overlay of bright-field and fluorescence images were taken 48 hours post-infection. The number of CD46 molecules/cell is indicated for each cell line. NA, not applicable (i.e., the anti-human CD46 antibody does not cross-react with African green monkey kidney [Vero] cells). Different MOI was used for visualization purposes. **(B)** Flow-cytometric analysis of the cell surface expression of MeV-receptors on B95m cells after incubation with PE-specific antibodies. The determined number of molecules per cell is indicated in brackets. **(C)** Quantification of infection of cell lines with MeV and MeV--Stealth. The data are shown as the mean ± SD (n = 3). All data were analyzed 48 hours post-infection and are normalized to the respective infectivity of the indicated virus in Vero-αHIS. Significance was determined using two-way ANOVA with Turkey's multiple comparison test. Ns, non-significant; $p > 0.05$; *, $p < 0.0366$; ****, $p < 0.0001$. Scale bar, 1 mm.
(TIF)

**S9 Fig. Lack of cross-neutralization between measles virus and Stealth. (A)** Virus neutralization assay for MeV and Stealth. Human AB pooled serum (left panel) or ferret anti-CDV serum (right panel) was used. Relative infection refers to the amount of infection in the presence of serum compared with that in the absence of serum. Values were calculated from two or three biological replicates performed in technical quadruplicates. (B) Antisera from infected Ifnar[tm]-CD46Ge mice was also used to determine the cross-neutralization between the viruses, n = 8 (note that some data points overlap). ND50 titers were converted to mIU/mL based on the ND50 obtained for MeV when assessed with the 3[rd] WHO International serum standard (3 IU/mL).
(TIF)

**S1 Data.**
(XLSX)

## Acknowledgments

We thank Dr. Isabel Simarro for providing tissue samples from dogs with distemper; Dr. Jason Moffat for providing the sequences of anti-CD46 scFv fragments; Dr. Veronika Von Messling for providing the anti-Fcyt antibody. Dr. M. Cristine Charlesworth in the Mayo Clinic Medical Genome Facility-Proteomics Core for size exclusion analysis; the Histology Core Laboratory at the Mayo Clinic Arizona for paraffin embedding and sample sectioning.

## Author Contributions

**Conceptualization:** Miguel Ángel Muñoz-Alía, Stephen J. Russell.

**Data curation:** Miguel Ángel Muñoz-Alía.

**Formal analysis:** Miguel Ángel Muñoz-Alía, Alexander Tischer.

**Funding acquisition:** Stephen J. Russell.

**Investigation:** Miguel Ángel Muñoz-Alía, Rebecca A. Nace, Alexander Tischer, Lianwen Zhang.

**Methodology:** Miguel Ángel Muñoz-Alía, Eugene S. Bah.

**Project administration:** Stephen J. Russell.

**Resources:** Miguel Ángel Muñoz-Alía, Matthew Auton, Stephen J. Russell.

**Software:** Miguel Ángel Muñoz-Alía.

**Supervision:** Stephen J. Russell.

**Validation:** Miguel Ángel Muñoz-Alía, Stephen J. Russell.

**Visualization:** Miguel Ángel Muñoz-Alía.

**Writing – original draft:** Miguel Ángel Muñoz-Alía.

**Writing – review & editing:** Miguel Ángel Muñoz-Alía, Stephen J. Russell.

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
