## [Decision Letter · Decision Letter 0]

21 Jul 2020

Dear Dr Muñoz-Alía,

Thank you very much for submitting your manuscript "MeV-Stealth: A CD46-specific Oncolytic Measles Virus Resistant to Neutralization by Measles-Immune Human Serum" for consideration at PLOS Pathogens. As with all papers reviewed by the journal, your manuscript was reviewed by members of the editorial board and by several independent reviewers. In light of the reviews (below this email), we would like to invite the resubmission of a significantly-revised version that takes into account the reviewers' comments.

We cannot make any decision about publication until we have seen the revised manuscript and your response to the reviewers' comments. Your revised manuscript is also likely to be sent to reviewers for further evaluation.

Sincerely,

Alexander Bukreyev, Ph.D.

Associate Editor

PLOS Pathogens

Christopher Basler

Section Editor

PLOS Pathogens

Kasturi Haldar

Editor-in-Chief

PLOS Pathogens

orcid.org/0000-0001-5065-158X

Michael Malim

Editor-in-Chief

PLOS Pathogens

orcid.org/0000-0002-7699-2064

Reviewer's Responses to Questions

**Part I - Summary**

Reviewer #1: The Russell laboratory, one of the most authoritative in the OV field, pioneered the field of MeV retargeting to cancer-specific receptors (ref 26). Here, Muñoz-Alía et al report on a strategy for MeV retargeting to CD46 which also confers evasion from human anti-MeV antibodies. The retargeting principle is based on swapping of MeV glycoproteins with the CDV homologs, and retargeting to CD46 by scFv insertion.

The initial experiments were carried out to select a Canine Distemper Virus (CDV) strain capable of using CD-46 and of high fusogenic activity towards CD46+ cells, as well as towards cells positive for nectin4 or SLAMF1.

The assay for selection of the best CDV strain and of the best combination of N and F was also based on a cell-cell fusion assay. Based on the same assay, the Authors then selected a scFv to CD46, and, furthermore, introduced single amino acid substitutions for detargeting from nectin4 and from SLAMF1.

The Authors then constructed the MeV-Stealth virus. When tested against a panel of CHO derivatives Stealth is retargeted to Cd46, detergeted from nectin4, hSLAM, but not from dog SLAM.

The Stealth virus was then assayed for oncolytic activity against SK-OV-3 cells implanted in immunodeficient SCID mice. Some of the mice received passive immunization with human anti-MeV Abs. The Stealth virus reduced tumor growth more to higher extent than MeV. The anti-MeV inhibited the Mev-mediated reduction to higher extent than the Stealth-mediated reduction.

In the Discussion, the Authors foresee that in an immunocompent mouse model the T-cell response would much aid the antitumor efficacy of the recombinants. A conclusion I agree with.

The work is well conducted. The conclusions are supported by results. The paper is clearly written.

Reviewer #2: Considerable effort has been made in development of measles virus (MeV) as an oncolytic agent for treatment of cancer. However, pre-existing immunity in humans limits its efficacy. A previous report showed that substitution of the related canine distemper virus (CDV) envelope proteins avoided the effects of neutralizing antibodies. However, this virus has undesirable receptor specificities, which the present study addresses by engineering CDV H protein to be specific for CD46 (the receptor for attenuated strains of MeV, that is overexpressed on many cancers) and to lack binding to human Nectin-4. The approaches for re-targeting by fusing sFv fragments to the H protein and mutating its normal receptor binding sites have been developed previously. However, achieving the desired result would be a significant advance. The strength of the study consists of preliminary evidence for the desired specificity in the new virus (Fig. 4C) and its oncolytic activity in vivo in the presence of passive immunization with anti-MeV serum (Fig. 5 D or E [mislabeled]). The weaknesses of the study are that the manuscript is not clearly written and most of the data need major modifications.

Reviewer #3: Steve Russel and colleagues present a nice study to show that a stealthed form of the measles OV MeV is capable of resisting neutralization by human serum antibody (passive immunization) while retaining the ability to slow the growth the growth of ovarian (orthotopic) and myeloma (xenograph) tumor models in mice. The “stealth” vector was produced by pseudotyping the MeV strain with glycoproteins (H/F) from a related but antigenically distinct virus, canine distemper virus (CDV). Vector retargeting was achieved by fusing CDV-H to a CD46 specific single chain antibody fragment and the F protein was modified to avoid recognition of nectin-4 and functioning well with the retargeted H glycoprotein. The tumor models expressed CD46. This is a good approach to avoiding initial virus inactivation during systemic delivery. An interesting finding was that the affinity for the targeted receptor played a role in fusogenic activity.

Comments:

1. Can the stealth vector be used in repeat dosing experiments or does the animal create new neutralizing antibody. This is of interest since many vectors will require more than one administration. Could be done in immune competent mice rather than tumor models.

2. While the mouse system did not allow evaluaton of anti-tumor immunity, can this be done with humanized NSG mice and xenographs. The lack of replication in mice also makes it difficult to evaluate safety. Do the authors believe that the new vector envelope constructs might induce anti-tumor immunity comparable to the MeV strain. Since different surface glycoproteins are in play the nature of immune cell recruitment could be different as well as abscopal activity in an immune responsive animal system.

3. Is the stealth vector effective if targeted to nectin-4 that is likely to be present on many tumor types.

4. Since tumor killing is not complete, are resistant cell types selected that lack target receptor or is this a matter of insufficient replication and vector spread.

5. IN Fig.4, why does the stealth virus replicate in dogSLAMF1 cells?

6. Is the stealth vector equally sensitive to innate tumor immune responses as MeV.

7. What role does ADCC play in curbing MeV activity-is it just VN.

8. Is there a difference in stealth virus VN titers in the absence and presence of complement. What role does complement play in virus neutralization.

9. While the authors show that in fig 5 D, the stealth virus is resistant to VN there is a small difference in VN treatment and untreated although not significant. Wondering if some epitopes are shared with MeV that will block replication. Nevertheless Suppl fig 6 is convincing.

**Part II – Major Issues: Key Experiments Required for Acceptance**

Reviewer #1: 1. My major concern regards the genetic stability of nectin4- and SLAM- detargeting through single amino acids substitutions. How stable is the virus if serially passaged (let’s say 10 times) in human cancer cells simultaneously positive for CD446, nectin4 and SLAM?. Would some of the virus revert to nectin4 or SLAM usage?

2. On a similar line of reasoning, to what extent is the Stealth virus really retargeted/detargeted? verification of tropism retargeting (fig. 4) should be performed also on human cells deficient for one or the other of the 3 receptors.

3. The Authors refer to avidity and affinity of the antibodies and/or retargeting moieties. None of the assay that the Authors applied (co-immunoprecipitation, ELISA) actually measures affinity of binding. The sentences should be rephrased, to avoid avidity, affinities, etc.

4. Fig. 4 B. the results are affected by the affinity of the antibodies being employed for detection. The Authors should in addition provide a PFU/genome copy number determination.

5. Fig. 4 A. There is some background virus growth in Vero-His. Can this virus be serially passaged? and what would be the receptor usage for the passaged virus: is the retargeting somehow leaky?

6. The Authors quote a very interesting paper from Zamarin group (Ref 55) which showed that some extent of prior immunity can be also beneficial. In this case, for example, what would be the effect of a prior T-cell immunity, as opposed to passive immunization with Antibodies? I would welcome a discussion in which the Authors address different facets of prior immunity, and the limits of passive immunization with Abs only.

Reviewer #2: 1. Most of the micrographs are not publication quality and are not readily interpretable in their current form. This includes Fig. 1A and B, Fig. 2A, Fig. 4C, and S5.

2. To the authors' credit, they do use one of (several) quantitative assays for cell-to-cell fusion mediated by cotransfection of H and F plasmids (Fig. 2B and D, Fig. 3B), although the basis for this one is not clearly described. However, statistical tests of one representative experiment performed in triplicate is not adequate to establish reproducibility. I know that normalization between luminescence experiments can be challenging, but typically involves comparison with a control.

3. Flow cytometry data in Fig. S2 do not support the conclusion that the glycoproteins in transfected cells are expressed at similar levels. This requires analysis by mean fluorescence intensity, not percent transfected cells, which I presume to be represented (it is not stated). Also need repeats and statistics here as well.

4. Better quantification of the ability of the new recombinant virus to replicate in cells that express different receptors in Fig. 4 is required to conclude that "CD46-retargeted CDV envelope glycoproteins determine virus tropism."

5. Fig. 5 does not present compelling evidence for the effectiveness of the new virus. Typically 5 mice are not sufficient to establish this. The luminescence images (Fig. 5D?) are not readily interpretable when presented like this.

Reviewer #3: no specific experiments

**Part III – Minor Issues: Editorial and Data Presentation Modifications**

Reviewer #1: 7. Fig. 5. lettering of panels in text is confused. Please, correct

8. Line 778, please correct.

Reviewer #2: 1. The Results section would benefit from a clearer presentation. I suggest the following for each experiment: 1) a clear statement of the purpose of the experiment, 2) enough explanation of the experimental approach for the reader to understand what was done and how the data are presented, 3) which results support the conclusion. In most cases, only the conclusion is stated.

2. A related problem is that the figure legends need to be sufficiently complete that the figure can be understood without reference to the text. This tends to overlap slightly with the second point above, and with the Materials and Methods section, but better that than the insufficient information provided.

3. Minor edits are needed for correctness: cell-to-cell fusion is not the same as virus envelope fusion. This can be seen by the fact that even H/F combinations with little activity seem to function fine in the viruses from which they were derived. The cell-to-cell fusion assay is fine to test the effects of mutations, etc. Also, co-immunoprecipitation is not a measure of affinity or avidity.

4. There are a number of misspellings and careless errors.

Reviewer #3: the review has a number of points that should be addressed by the authors.

PLOS authors have the option to publish the peer review history of their article (what does this mean?). If published, this will include your full peer review and any attached files.

Reviewer #1: **Yes: **Gabriella Campadelli-Fiume

Reviewer #2: No

Reviewer #3: No
---

## [Decision Letter · Decision Letter 1]

22 Nov 2020

Dear Dr Muñoz-Alía,

Thank you very much for submitting your manuscript "MeV-Stealth: A CD46-specific Oncolytic Measles Virus Resistant to Neutralization by Measles-Immune Human Serum" for consideration at PLOS Pathogens. As with all papers reviewed by the journal, your manuscript was reviewed by members of the editorial board and by several independent reviewers. The reviewers appreciated the attention to an important topic. Based on the reviews, we are likely to accept this manuscript for publication, providing that you modify the manuscript according to the review recommendations.

As you can see, two of the reviewers for the revised version of the manuscript were not reviewers of the first submission. Each raises additional concerns that should at least be addressed in the text of an updated version. Experimentally, you may consider including studies with human cells to address virus tropism. This is a major concern of Reviewer 4.

Sincerely,

Christopher Basler

Section Editor

PLOS Pathogens

Kasturi Haldar

Editor-in-Chief

PLOS Pathogens

orcid.org/0000-0001-5065-158X

Michael Malim

Editor-in-Chief

PLOS Pathogens

orcid.org/0000-0002-7699-2064

Reviewer Comments (if any, and for reference):

Reviewer's Responses to Questions

**Part I - Summary**

Reviewer #1: (No Response)

Reviewer #4: This is an interesting manuscript describing a novel clinically promising oncolytic MEV-CDV chimeric virus from a top group in the field of oncolytic virotherapy. It should be noted that multiple previous studies described various MEV-CDV chimeras. The current manuscript presents the following new data:

1. The novel engineered oncolytic MEV-CDV chimeric virus (“MeV-Stealth”) specifically targets CD46-expressing cells.

2. The MeV-Stealth virus has diminished undesirable tropism for human NECTIN-4 and hSLAMF1

3. The antitumor efficacy of MeV-Stealth in myeloma and ovarian tumor-bearing mice was similar to that of vaccine-lineage MeV.

4. Unlike MV, treatment of ovarian tumors with MeV-Stealth significantly increased overall survival (compared with treatment with vaccine-lineage MeV).

Although the revised manuscript describes a lot of data and addresses many of the original critical points, there are still several important major deficiencies that I describe in Part II.

Reviewer #5: In the manuscript, entitled “MeV-Stealth: A CD46-specific Oncolytic Measles Virus Resistant to Neutralization by Measles-Immune Human Serum”, Muñoz-Alía et. al. show that a measles virus variant, containing an envelope glycoprotein from a canine distemper virus further modified to retarget to CD46, is capable of evading detection by neutralizing antibodies, present in human serum. The authors selected their stealth combination (CDV-H 5804/F SPA) based on higher fusion ability and showed that scFv-CD46 addition indeed direct the virus towards CD46 and not NECTIN-4.

**Part II – Major Issues: Key Experiments Required for Acceptance**

Reviewer #1: The Authors have satisfactorily addresses all the issues and concerns raised by this reviewer.

Reviewer #4: Although the revised manuscript describes a lot of data and addresses many of the original critical points, there are still several important major deficiencies:

1. RETARGETTING. The manuscript present a large amount of data demonstrating targeting of MeV-Stealth to CD-46-expressing cells and de-targeting from human NECTIN-4 and hSLAMF1. Unfortunately, practically all the data are from various CHO cell lines expressing (or not) these genes. In fact, the originally submitted manuscript was criticized for the lack of proof using various human cells lines expressing or not CD46, NECTIN-4 and hSLAMF1. The authors responded that “Chinese Hamster Ovarian cells (CHO) expressing individually each of the different receptors are used as the gold standard to determine virus tropism. To fulfill this reviewer’s request, we have included Figs 4B, 4C; Fig. 5 and Supplementary Figure 7…”. I do not agree with this response, and these figures do not show the requested data. CHO cells are very useful as a screening tool, but, ultimately, the novel tropism should be confirmed in a panel of non-malignant and malignant cell lines expressing one of these 3 proteins. Importantly, the expression levels and PTMs for these proteins could be very different in human and CHO cells, which could affect tropism in human vs CHO cells or human cancer vs human non-malignant cells. Interestingly, the same authors recently published a similar paper in Molecular Cancer Therapy describing CD38-targetted MEV-CDN chimeric virus (PMID: 32847970: “Retargeted and Stealth-Modified Oncolytic Measles Viruses for Systemic Cancer Therapy in Measles Immune Patients, Molecular Cancer Therapy 2020 Oct;19(10):2057-2067. doi: 10.1158/1535-7163.MCT-20-0134), in which they actually conducted this critical experiment (Fig. 3C: “Chimeric MV bearing CDV F and CDV H retargeted to EGFR or CD38 maintains receptor-specific infectivity in selected human tumor cell lines that express either EGFR or CD38.”)

2. ONCOSELECTIVITY. Testing MeV-Stealth (and comparing it to the MV and at least some of the previously described MeV-CDN chimeras) in a panel of normal and malignant human cell lines is also important to demonstrate oncoselectivity of this novel virus MeV-Stealth. Due to the mentioned in the manuscript limitations of the in vivo murine system (MeV does not replicate in mouse cells), this critical question is not addressed in this manuscript.

3. GENERAL EFFICACY IN VIVO. The demonstrated similar antitumor efficacy of MeV-Stealth to that of vaccine-lineage MeV is not so novel and was previously shown for other MeV-CDN chimeras.

5. EFFICACY IN VIVO IN THE RESENCE OF MEV AB. I am a little bit puzzled why MeV-Stealth antitumor efficacy was so clearly inhibited by MeV antibody (Fig. 7B)? Also, how did authors chose the dose of the ab? I noticed that the current manuscript used “600 mIU of anti-MeV IgG antibody intraperitoneally three hours before virus treatment”, while the mentioned above Molecular Cancer Therapy paper describing CD38-targetted MEV-CDN chimeric virus (PMID: 32847970) indicated “60 EU of pooled human measles-immune neutralizing serum added 3 hours before virus treatment”. Am I correct that 100 times less MeV ab was used in the current study? Why?

Reviewer #5: One weakness in the story (to my opinion) concerns the method used to indicate particle-to-pfu ratios of the MeV-Stealth virus compared to the original MeV (lines 193-195, and lines 279 -280 in discussion) and Figure 4E. Although it could well be that there is indeed no difference in the incorporation of the foreign envelop in the Stealth virus, a western blot is not the correct experiment to show that for the following reasons:

- The authors do not use purified viruses, since they use virus stocks, freed from cells by freeze-thawing. (Although in the material section this point is not directly clear, without looking up the reference mentioned) With a western blot, the authors cannot exclude that virus proteins produced by the cells are present in the cleared supernatant. Did does not necessarily mean that the proteins are incorporated in the virions. Furthermore, the amount of proteins in cell lysates can vary between different virus productions even when “equivalent numbers of particles” are used.

- Also, the authors state in line 195 that they observe “no significant difference” in expression levels of the major structural protein N. In fig 4E, however, there seems more N-protein present in the Stealth virus lane compared to the MeV lane. Without quantification it is very difficult to say something about significance (especially with only one western blot).

- For the stealth virus, the authors use an anti-His antibody and for the wt MeV a different antibody, recognizing MeV H-protein. It is well known that not all antibodies detect proteins with the same affinity on western blots. This makes a direct comparison very difficult.

If the authors want to make a conclusive point that the Stealth virus has a similar particle-to-PFU ratio they should show this with additional experiments and use more, (sucrose cushion) purified viruses or Electron Microscopy to quantify the number of particles in both (wtMeV and Stealth-Mev) supernatants with equal amounts of infectious virions. This would also give some information on the quality of the virions (presence of defective particles compared to intact virions).

**Part III – Minor Issues: Editorial and Data Presentation Modifications**

Reviewer #1: (No Response)

Reviewer #4: (No Response)

Reviewer #5: 1. Line 128, “the affinity of CDV-F SPA for CDV-H 5804 was slightly lower than that of CDV-F SPA for CDV-H SPA (Figure 2C)” Based on figure 2C alone, without any quantification, it is difficult to see a difference in intensity between SPA(M437) and 5804. Could the authors re-phrase this sentence or delete it and leave the interpretation up to the readers.

2. Line 169, “(Supplementary Figure 5)”, I think this should be changed to Supplementary Figure 6

3. Line 188, “(Figure 4C, Supplementary Figure 6)” This should be changed to Supplementary Figure 7 (I think)

4. Line 243, “our results, presented in Supplementary Figure 7”. This should be changed to Supplementary Figure 8.

5. Materials and Methods, section “SDS-PAGE and immunoblotting”

Lines 453-54, “Band quantification was carried out using the KwikQuant Image Analyzer 1.4. (Cat. # D1016, Kindle Biosciences, LLC)” I could not find any quantification data in the manuscript, only speculations; see comment with line 128. If it is not used in the manuscript, please remove the sentence or provide the data.

6. Figure 3B. It is difficult to distinguish the black lines from the red lines

7. Line 824, “Figure 4. CD46-retargeted CDV envelope glycoproteins determine virus tropism. (D) Schematic” D should be changed to A

8. Line 840 also Figure 4; panel G. “Representative microphotographs are shown after infection for three or six days”. Figure only shows pictures from Day 3, but not day 6, as is stated.

9. Supplementary Figure 4, panel B. Typo: Volumen instead of Volume.

PLOS authors have the option to publish the peer review history of their article (what does this mean?). If published, this will include your full peer review and any attached files.

Reviewer #1: **Yes: **Gabriella Campadelli-Fiume

Reviewer #4: No

Reviewer #5: No
---

## [Editor Report · Decision Letter 2]

5 Jan 2021

Dear Dr Muñoz-Alía,

We are pleased to inform you that your manuscript 'MeV-Stealth: A CD46-specific Oncolytic Measles Virus Resistant to Neutralization by Measles-Immune Human Serum' has been provisionally accepted for publication in PLOS Pathogens.

Best regards,

Alexander Bukreyev, Ph.D.

Associate Editor

PLOS Pathogens

Christopher Basler

Section Editor

PLOS Pathogens

Kasturi Haldar

Editor-in-Chief

PLOS Pathogens

orcid.org/0000-0001-5065-158X

Michael Malim

Editor-in-Chief

PLOS Pathogens

orcid.org/0000-0002-7699-2064
---

## [Editor Report · Acceptance letter]

28 Jan 2021

Dear Dr Muñoz-Alía,

We are delighted to inform you that your manuscript, "MeV-Stealth: A CD46-specific Oncolytic Measles Virus Resistant to Neutralization by Measles-Immune Human Serum," has been formally accepted for publication in PLOS Pathogens.

Best regards,

Kasturi Haldar

Editor-in-Chief

PLOS Pathogens

orcid.org/0000-0001-5065-158X

Michael Malim

Editor-in-Chief

PLOS Pathogens

orcid.org/0000-0002-7699-2064